# DON'T PAY ATTENTION

## ABSTRACT

The Transformer has become the de facto standard for modern language models owing to its parallelizable training and effective autoregressive decoding. However, its fixed context window and the quadratic time and memory costs of its self-attention mechanism remain central bottlenecks. These constraints have revived interest in recurrent architectures that scale linearly with sequence length, but at the cost of reduced parallelism. In this paper, we introduce Avey, a new foundational architecture that breaks away from both attention and recurrence. Avey pairs a ranker with an autoregressive neural processor to select and contextualize only the most relevant tokens for any given token. Specifically, it decouples sequence length from context width, thus enabling effective and efficient processing of arbitrarily long sequences. Results show that Avey compares favorably to the Transformer across a variety of standard short-range NLP benchmarks, while significantly outperforming it on tasks requiring long-range dependency modeling.

## 1 INTRODUCTION

The Transformer (Vaswani et al., 2017) has emerged as one of the most influential AI innovations in recent years, profoundly impacting various aspects of modern life, including work, science, and art, to mention just a few. Notably, Large Language Models (LLMs) are almost universally based on the Transformer (Gu and Dao, 2023), which has demonstrated remarkable performance in natural language processing (NLP) (Ouyang et al., 2022; Liu et al., 2019; Raffel et al., 2020) and various other fields (He et al., 2022; Liu et al., 2021b; Baevski et al., 2020).

The Transformer's state-of-the-art performance is largely driven by its recurrence-free self-attention mechanism, which enables parallel processing of entire token sequences. Nevertheless, the computational and memory costs of self-attention grow quadratically with sequence length, making it inefficient for handling arbitrarily long sequences. Extensive research has been conducted over the years to address this limitation (Tay et al., 2022), with a noticeable emphasis on linearizing attention (Katharopoulos et al., 2020; Choromanski et al., 2020; Zhai et al., 2021; Wang et al., 2020). These linear approaches aim at approximating self-attention in a more computationally efficient manner, without considerably compromising performance.

Nonetheless, linear attention mechanisms have generally underperformed the original self-attention mechanism, often by a significant margin in language modeling tasks (Yang et al., 2023; Kasai et al., 2021). While recent linear models such as RWKV (Peng et al., 2025) and RetNet (Sun et al., 2023) have shown promising results, substantial progress is still needed before they can reliably and consistently surpass the Transformer (Li et al., 2024b). In addition, these models have yet to demonstrate definitive empirical effectiveness at scale (Gu and Dao, 2023). This persistent performance gap between quadratic and linear approaches has spurred renewed interest in RNN-based architectures, which offer linear scalability with sequence length but limit parallelism due to their inherently cyclical nature.

To exemplify, state space models (SSMs) (Kalman, 1960; Gu et al., 2021b), which are viewed as extensions of RNNs, have recently emerged as a compelling class of architectures. Unlike traditional RNNs, SSMs can parameterize their state transition matrices in a structured manner (e.g., via using a diagonal plus low-rank decomposition) to improve computational efficiency and enhance gradient flow. A specialized subclass of these models, known as structured state space sequence (S4) models (Gu et al., 2021a;b), has garnered growing attention. Yet, despite their theoretical ap-

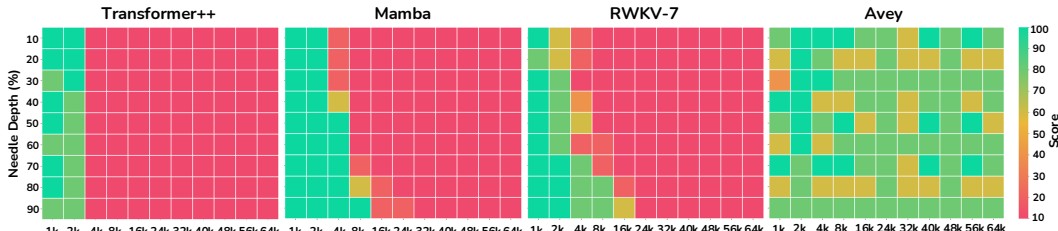

Figure 1: Needle-in-a-Haystack test performance comparison between Transformer++, Mamba, RWKV-7, and Avey, all using 1.5B parameters. The x-axis denotes the lengths of haystacks (i.e., documents with distractor texts, varying from 2k to 64k tokens) and the y-axis refers to the position of the needle (i.e., a short sentence) within any of the haystacks. A green cell indicates successful needle recall, while a red cell indicates failure. Transformer++, Mamba, and RWKV-7 were trained with 2k-token context windows, whereas Avey was trained with *only* a 512-token window yet was able to extrapolate to the longest sequences evaluated.

peal, S4 models struggled with language modeling tasks, typically trailing Transformers by several points (Fu et al., 2022; Gu et al., 2021a).

Most recently, Mamba (Gu and Dao, 2023) advanced S4 models by enhancing their selectivity and effectiveness while enabling high training concurrency. It demonstrated strong performance on tasks involving long-range dependencies and compared favorably to Transformers in language modeling. However, training, scaling, and interpreting Mamba—and SSMs more broadly (Smith et al., 2022; Poli et al., 2023; Hasani et al., 2022)—remain challenging, while continue to be promising (Dao and Gu, 2024).

We posit that the primary limitation of the Transformer lies in its inability to effectively model dependencies beyond its fixed context window. While its core self-attention mechanism is inherently parallelizable, this constraint makes its quadratic complexity a significant bottleneck *at scale*. This explains the surge of research aimed at reducing this complexity or exploring RNN-inspired alternatives. In this work, we propose a more viable approach by *decoupling* context width from sequence length, allowing models to scale to arbitrarily long sequences. Under this paradigm shift, the quadratic training complexity becomes less of a concern when small context windows are used, especially if the models maintain high parallelizability.

This paper introduces **Avey**[1], a new architecture for language modeling that departs from Transformer-based and RNN-like designs. Avey is a flexible, sequence-length-invariant model that decouples sequence length from context width, thus enabling effective processing of long-range sequences. It preserves the influence of tokens that appear *outside* its context window, regardless of their positions in the sequence. This is achieved via a weighted-selective-split interaction mechanism, which systematically skips irrelevant tokens beyond the context window and ensures direct interactions with only relevant ones, retaining their contributions irrespective of sequence length.

Fig. 1 illustrates Avey's ability to generalize beyond its training context. A popular benchmark for evaluating this capability is Needle-in-a-Haystack (NiaH) (Kamradt, 2023). This benchmark measures a model's capacity to recitep a specific sentence (i.e., the *needle*) placed at an arbitrary position within a large body of distractor text (i.e., the *haystack*). Since its introduction, NiaH has become a widely used sandbox for probing the limits of long-context language models in capturing distant dependencies, and smaller models in generalizing beyond their trained context windows (Fu et al., 2024). As shown in the figure, Transformer++ (i.e., the Transformer with an enhanced architecture and training recipe– see Section 3.1), which was trained with a 2k-token context window, could not generalize beyond that limit. In contrast, Mamba and RWKV-7 (Peng et al., 2025), also trained with 2k-token windows, managed to generalize to nearly 8k and 16k tokens, respectively. Most notably, Avey, despite being trained on a context window of *only* 512 tokens, successfully generalized to the maximum tested sequence length of 64k tokens, demonstrating strong extrapolative capability far beyond its original training regime.

---

[1]Avey is not an acronym, but a name that the authors like.

To elaborate on its technical aspects, Avey is a recurrence- and attention-free architecture comprising two principal components, a ranker and a neural processor. The ranker slices each input sequence into splits of consecutive tokens and selects the top $k$ most relevant splits for each current split being processed by the neural processor. The neural processor consists of three core units, the enricher, contextualizer, and fuser. The enricher enhances the quality of token embeddings by expanding their learnable features using a position-wise neural network. The contextualizer is an embedding-wise neural network with dynamic parameterization, enabling interactions between relevant tokens across the current and top $k$ splits. Lastly, the fuser learns a function that integrates the contextualized features produced by the contextualizer with some uncontextualized features bypassed by a partial-embedding bypassing mechanism.

To summarize, our main contributions in this paper are as follows:

- We propose Avey, a new recurrence- and attention-free neural architecture that decouples context window from sequence length, thus enabling effective processing of long-range sequences.

- We show that Avey performs comparably to the Transformer—outperforming it at two model sizes and underperforming it at one—across a range of popular zero-shot NLP benchmarks, thereby establishing an initial foundational architecture with potential for more scalable and effective language modeling.

- In contrast to the Transformer, we demonstrate that Avey can scale far beyond its context window using the standard Single Needle-In-A-Haystack (S-NIAH) benchmark suite from RULER (Hsieh et al., 2024).

- We show that Mamba (representing SSMs) and RWKV-7 (representing linear attention models) exhibit some ability to generalize beyond their training context windows, but their performance decline significantly as the sequence length increases far beyond them. By comparison, Avey consistently and substantially outperforms both Mamba and RWKV-7 on the S-NIAH benchmark suite.

- We conduct extensive ablation studies to assess the impact of each design choice in Avey.

- We provide a comprehensive survey of related work in Appendix O.

- We open-source the code and pretrained checkpoints of Avey to facilitate reproducibility and foster future research (see Section 6).

## 2 AVEY

As indicated earlier, Avey comprises two components, a ranker and a neural processor. We next describe each component in detail (see Appendix R for more design intuitions behind them).

### 2.1 RANKER

Avey decouples sequence length from context width, enabling the processing of arbitrarily long sequences. The sequence length refers to the total number of tokens in a sequence, while the context width denotes the number of tokens that the neural processor can contextualize simultaneously. Importantly, the sequence length can be set to a value that is much larger than the context width. As such, the influence of *global* tokens (or tokens that fall outside the context window) may diminish as more tokens are successively processed. If such global tokens are semantically relevant to *local* tokens (or tokens that fall within the context window), the quality of token representations will decline, and the effectiveness of the model will degrade.

To this end, the ranker and neural processor jointly employ a **weighted-selective-split interaction mechanism**, which skips irrelevant global tokens and ensures direct interactions with relevant ones, preserving their impact regardless of sequence length. As demonstrated in Fig. 2, Avey divides each input sequence into equal-sized *splits*, each consisting of a list of contiguous token embeddings. Prior to predicting the next token in the sequence (e.g., token 9 in the figure), Avey involves the ranker to identify the top $k$ (e.g., 2 in the figure) splits that are most relevant (e.g., splits 1 and 3) to the *current split* (i.e., split 4).

The current split is defined as the one that either contains the token to be predicted (e.g., split 4 may contain only embedding 7, and Avey will aim to predict token 8) or contributes to predicting the *first*

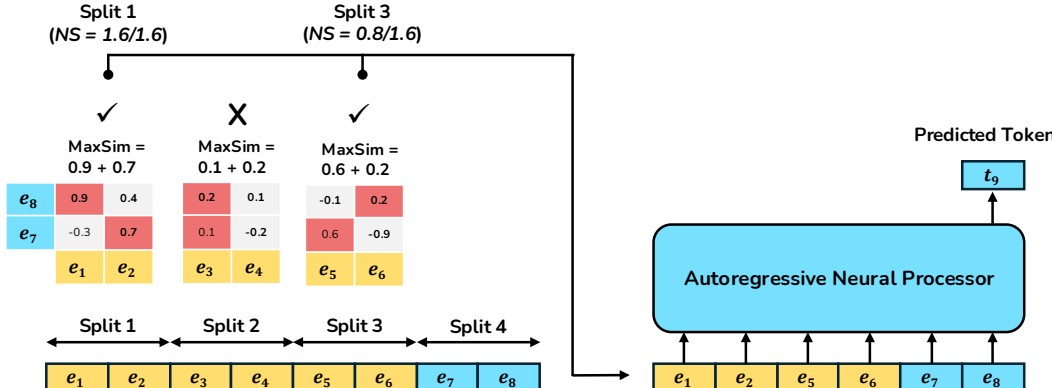

Figure 2: The ranker (left) partitions each input sequence into equal-sized splits and identifies the top $k$ most relevant ones (e.g., splits 1 and 3 for $k = 2$) with respect to the *current split* (e.g., split 4), using the MaxSim operator. These top-$k$ splits are then weighted by their normalized scores, where the normalized score (NS) of a split is computed as the ratio of its MaxSim value to the highest MaxSim score among the $k$ splits. Finally, the weighted top-$k$ splits are contextualized together with the current split by the neural processor (right).

token in its subsequent split (e.g., split 4 serves in predicting token 9, which will belong to split 5 once predicted). To determine relevance, the ranker computes a similarity score between the current split, say $S_c$, and each preceding split, say $S_p$, using the MaxSim operator (Khattab and Zaharia, 2020), originally proposed and utilized in Information Retrieval. Specifically, pairwise similarities are calculated (e.g., using a cosine function) between each embedding in $S_c$ and all embeddings in $S_p$. For each embedding in $S_c$, the maximum similarity across all $S_p$'s embeddings is taken, and then the maxima of all $S_c$'s embeddings are added to yield the final MaxSim score (see Fig. 2). This score signifies how relevant $S_p$ is to $S_c$.

Subsequently, the preceding splits are ranked based on their MaxSim scores, and the top $k$ most relevant ones are contextualized with the current split by the neural processor, while maintaining their original order in the sequence. Before contextualization, however, the MaxSim scores of the top $k$ splits are normalized with respect to the highest MaxSim score among them (e.g., split 3's MaxSim score of 0.8 in Fig. 2 is normalized via dividing it by the maximum score among the top $k$ splits, i.e., 1.6, yielding 0.5). Each selected split is then *weighted* by its corresponding normalized MaxSim score, effectively scaling its contribution during contextualization.

As a result, the weighted-selective-split interaction mechanism does not only allow the ranker to rank splits based on relevance but also the neural processor to contextualize them accordingly, as each selected split is pre-weighted by its relevance score. This empowers the neural processor to judiciously leverage global information (i.e., splits beyond the context width) by focusing selectively on only relevant features, emphasizing informative ones and deemphasizing less useful ones, thus enhancing performance. We analyze the impact of weighting the top $k$ splits in Appendix K.

Note that the ranker is invoked *only once* per full forward and backward passes[2]. To elucidate, Avey's depth can be increased by stacking multiple layers within the neural processor (see Fig. 3), thereby enabling the modeling of complex, hierarchical patterns. In contrast, only a single ranker is required before the stack of layers within the processor, regardless of their number. Once the ranker identifies the top $k$ relevant splits of the current split, the neural processor contextualizes them all using one or more layers.

Consequently, during training, each current split is matched once against every preceding split. This results in a compute cost of $\frac{N/S \cdot (N/S+1)}{2} \cdot S^2 d$ or a time complexity of $O(N^2 d)$, where $N$ is the sequence length, $S$ is the split size, and $d$ is the embedding dimension. This complexity assumes

---

[2]It is important to note as well that the ranker is an *internal* module that selects among *in-sequence* splits already present in the input. It does not query external corpora or indexes and therefore adds neither retrieval latency nor corpus-freshness dependencies. It is *not* a RAG component, which fetches *external* evidence at inference (and/or training) time from a separate knowledge base. The two are orthogonal indeed (one allocates internal context and the other changes the evidence set) and can be composed. See Appendix P for details.

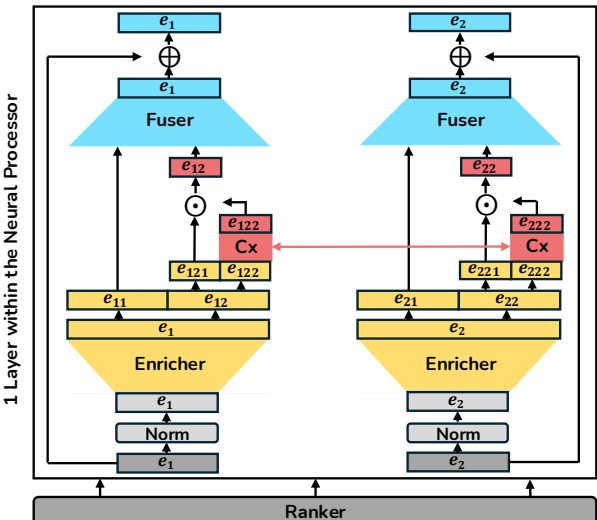

Figure 3: The neural processor (top) with its three major components, the enricher, contextualizer (Cx), and fuser. The processor is unfolded into two copies for illustrative purposes only, to show how different embeddings, (e.g., $e_1$ and $e_2$, or more precisely, parts of their tails, i.e., $e_{122}$ and $e_{222}$) are contextualized by Cx (i.e., in reality, all components are shared across all embeddings and many embeddings can be input to Cx simultaneously).

that scalar multiply-add operations (e.g., those used in computing cosine similarity for MaxSim) and comparisons (e.g., those utilized to determine maximum scores) are constant-time. We next discuss the neural processor.

## 2.2 NEURAL PROCESSOR

The neural processor encompasses three key machineries, the *enricher*, *contextualizer*, and *fuser* (see Fig. 3). We describe each in detail below.

### 2.2.1 THE ENRICHER

The enricher aims at enriching the quality of each token representation via expanding the quantity of its learnable features, thereby enabling the contextualizer to capture more nuanced distinctions between tokens. Concretely, it is a one-layer, position-wise neural network (i.e., the input to each neuron is a single scalar element from an embedding), thus operating on each embedding independently, without considering neighboring embeddings. As such, it allows *intra-feature interactions* within the context of each individual embedding, facilitating the learning of higher-order and more expressive representations. The enricher can expand each input embedding by an arbitrary factor. We study the effect of varying the expansion factor on Avey's performance in Appendix D, and ablate the enricher's contribution in Appendix K.

Equation 1 formalizes the enricher, where $\mathbf{X} \in \mathbb{R}^{C \times d}$ is a matrix of $C$ input embeddings ($C \leq N$, where $N$ is the sequence length), each of dimension $d$; $\sigma$ is an activation function; $\mathbf{U} \in \mathbb{R}^{d \times m}$ is a learnable weight matrix defining a linear projection from dimension $d$ to $m$, where $m > d$; and $\mathbf{b} \in \mathbb{R}^{C \times m}$ denotes biases.

$$\mathbf{Z} = \sigma(\mathbf{XU} + \mathbf{b}) \qquad (1)$$

As demonstrated in Fig. 3, the enricher feeds both, the contextualizer *and* the fuser. In particular, it bypasses a portion of each expanded embedding directly to the fuser in a technique that we refer to as **partial-embedding bypassing**. More precisely, the output of the enricher, $\mathbf{Z} \in \mathbb{R}^{C \times m}$, is split into two parts: (1) the *head* $\mathbf{Z}_h \in \mathbb{R}^{C \times m_h}$, which is bypassed directly to the fuser, and (2) the *tail* $\mathbf{Z}_t \in \mathbb{R}^{C \times m_t}$, which is forwarded to the contextualizer, where $m = m_h + m_t$. Consequently, varying the tail size alters the head size, which can influence Avey's performance. We investigate the impact of different tail sizes on Avey's performance in Appendix E.

The partial-embedding bypassing technique allows preserving raw distinctive features of each embedding, thus inducing representations with inherent diversity. This diversity may serve in alleviating issues like entropy collapse (Zhai et al., 2023), where the contextualizer increasingly focuses on a few tokens, and over-smoothing (Zhou et al., 2021; Shi et al., 2022; Zhou et al., 2024), where embeddings become increasingly similar, as Avey's depth is increased. We analyze the significance of partial-embedding bypassing on Avey's effectiveness in Appendix K.

Lastly, Equation 1 implies that each neuron performs a weighted sum of $d$ input features (i.e., the elements of an embedding), incurring $d - 1$ multiply-add operations. Since $d$ is projected to a higher dimension $m^3$, the total computational cost is $m(d - 1)$ per token. For a sequence of $N$ tokens, the cost becomes $Nm(d - 1)$, or asymptotically $\mathcal{O}(Nmd)$.

### 2.2.2 THE CONTEXTUALIZER

The contextualizer is a one-layer, embedding-wise neural network (i.e., the input to each neuron is one embedding), thus operating in parallel on $C$ embeddings, where $C$ denotes the context width. More precisely, it enables inter-embedding, *data-dependent* interactions of only tail embeddings (i.e., $\mathbf{Z}_t \in \mathbb{R}^{C \times m_t}$, as defined in Section 2.2.1), after each enricher's output embedding $m$ is split into a head part (i.e., $m_h$) and a tail part (i.e., $m_t$), and only the $m_t$ part (e.g., $e_{12}$ and $e_{22}$ in Fig. 3) is forwarded to the contextualizer.

The $m_t$ part of each enriched embedding is further divided into two equal portions, $m_{tl}$ (or left portion) and $m_{tr}$ (or right portion), to enable judicious control of information flow through the neural processor. Specifically, $m_{tl}$ serves as a *gating mechanism* for $m_{tr}$, regulating how much of its contextualized feature values are propagated forward. Both $m_{tl}$ and $m_{tr}$ are learnable by the model, hence, allowing $m_{tl}$ to dynamically capture the significance of each $m_{tr}$'s feature, and emphasize or deemphasize its influence accordingly. This gating mechanism was inspired from gMLP (Liu et al., 2021a) and resembles that of Gated Linear Units (Dauphin et al., 2017; Shazeer, 2020; Wu et al., 2019).

More formally, $\mathbf{Z}_t \in \mathbb{R}^{C \times m_t}$ is partitioned into two equal parts, $\mathbf{Z}_{tl} \in \mathbb{R}^{C \times (m_t/2)}$, which is bypassed to a multiplicative element-wise operation as part of a gating mechanism, and $\mathbf{Z}_{tr} \in \mathbb{R}^{C \times (m_t/2)}$, which is contextualized via a neural network, where each neuron takes as input an embedding of dimension $m_t/2$. Equation 2 defines the overall process, where $\mathbf{V} \in \mathbb{R}^{C \times C}$ is a learnable weight matrix representing a linear cross-embedding transformation, $\odot$ denotes element-wise multiplication, $\mathbf{b}' \in \mathbb{R}^{C \times (m_t/2)}$ refers to optional biases, and $\mathcal{N}(\mathbf{Z}_{tr})$ and $\mathcal{N}(\mathbf{Z}_{tr}^{\top})$ are row- and column-wise normalized versions of $\mathbf{Z}_{tr}$, respectively.

$$\mathbf{c}(\mathbf{Z}_t) = \mathbf{Z}_{tl} \odot \sigma \left( \left( \mathbf{V} \odot \mathcal{N}(\mathbf{Z}_{tr}) \mathcal{N}(\mathbf{Z}_{tr}^{\top}) \right) \mathbf{Z}_{tr} + \mathbf{b}' \right) \tag{2}$$

Equation 2 suggests that each neuron in the contextualizer's network performs a weighted sum of the cosine similarities between embeddings (denoted by $\mathcal{N}(\mathbf{Z}_{tr}) \mathcal{N}(\mathbf{Z}_{tr}^{\top})$) and the embeddings themselves (denoted by $\mathbf{Z}_{tr}$). This introduces a level of *selectivity* into the neural processor, as advocated by (Gu and Dao, 2023). Specifically, it makes the parametrization of the neural processor dynamic, enabling it to disregard or focus on information during inference based on the input. We examine the influence of dynamic parametrization on Avey's performance in Appendix K[4].

Finally, we note that the contextualizer inherently models the relationships between tokens, making the neural processor naturally aware of their positions in the sequence (i.e., positional encodings are not needed). In terms of complexity, as each neuron performs a weighted sum of $C$ embeddings, each of dimension $m_t/2$, it results in a cost of $(C - 1)m_t/2$ multiply-add operations. With $C$ neurons, the cost becomes $C(C - 1)m_t/2$. For a sequence of $N$ tokens, the contextualizer processes $N/S$ splits, each contextualized with $k$ relevant splits, making $C = S(k + 1)$ and yielding a total cost of $(N/S)[C(C - 1)m_t/2] = N(k + 1)[(C - 1)m_t/2]$ (after substituting $S$ with $C/(k + 1)$), or asymptotically $\mathcal{O}(NkCm_t)$.

---

[3]In our case, we experiment with $m$ being a multiple of $d$, entailing that $m \geq 2d$ (see Appendix D).

[4]See also a discussion on neural contextualization versus attention in Appendix Q.

### 2.2.3 THE FUSER

The fuser is designed to learn an optimal function, referred to as *fusion*[5], that integrates uncontextualized features (i.e., those of dimension $m_h$, bypassed by the partial-embedding bypassing technique) with contextualized features (i.e., those of dimension $m_t/2$, output by the contextualizer). Subsequently, it produces, for each input token, a *contracted* representation that matches the token's original embedding dimension $d$ (see Fig. 3). Akin to the enricher, it is a one-layer, position-wise neural network, which operates on each embedding of dimension $m_h + m_t/2$ independently.

Equation 3 provides a mathematical definition of the fuser, where $\mathbf{Z}_h \in \mathbb{R}^{C \times m_h}$ (as described in Section 2.2.1) and $\mathbf{c}(\mathbf{Z}_t) \in \mathbb{R}^{C \times (m_t/2)}$ (as suggested by Equation 2) are concatenated, and $\mathbf{O} \in \mathbb{R}^{(m_h + m_t/2) \times d}$ is a learnable weight matrix representing a linear projection from dimension $m_h + m_t/2$ back to dimension $d$, where $d < m_h + m_t/2$[6].

$$f(\mathbf{Z}) = [\mathbf{Z}_h \,\|\, \mathbf{c}(\mathbf{Z}_t)]\mathbf{O} \tag{3}$$

Equation 3 entails that each neuron performs a weighted sum of $m_h + m_t/2$ embedding elements, yielding a cost of $(m_h + m_t/2 - 1)$ multiply-add operations. For $d$ neurons (since the fuser projects $m_h + m_t/2$ to $d$), the cost is $d(m_h + m_t/2 - 1)$. For a sequence of $N$ tokens, the total cost is $Nd(m_h + m_t/2 - 1)$, or asymptotically $\mathcal{O}(Nmd)$.

Considering the aggregate computational costs of the ranker, enricher, contextualizer, and fuser, Avey exhibits a training time complexity of $\mathcal{O}(L(2Nmd + NkCm_t) + N^2d)$, where $L$ denotes the number of neural processor layers. As the term $N^2d$ dominates asymptotically, the overall complexity simplifies to $\mathcal{O}(N^2d)$. During inference, the complexity reduces to $\mathcal{O}(N)$, or linear per token. We elaborate on Avey's time complexity in Appendix N. In addition, we show that its empirical Time to First Token (TTFT), a key latency metric for real-time applications (Horton et al., 2024; Liu et al., 2025; Dexter et al., 2025), is significantly lower than that of Transformer++, Mamba, and RWKV-7 (see Fig. 8 in Appendix N).

## 3 EXPERIMENTS

### 3.1 EXPERIMENTAL SETUP

We compare Avey against three leading open-source models, namely, Mamba (Implementation, 2023a), RWKV-7 (Implementation, 2023b), and Transformer++, extended to the strongest architectural recipe of the standard Transformer (Karpathy, 2023) (see Appendix A for details). All models were trained using their best-known hyperparameters under a fixed budget of 100 billion tokens drawn from the FineWeb dataset (Hugging Face, 2023). Complete training and model hyperparameters for all the baselines are provided in Appendix A.

To assess each model's accuracy, we employed a suite of widely used NLP benchmarks, including ARC-E and ARC-C (Clark et al., 2018), HellaSwag (Zellers et al., 2019), PIQA (Bisk et al., 2020), OBQA (Mihaylov et al., 2018), SIQA (Sap et al., 2019), and Winogrande (Sakaguchi et al., 2021). Additionally, we evaluated the long-context retrieval capabilities of all the models using the standard Single Needle-In-A-Haystack (S-NIAH) benchmark suite from RULER (Hsieh et al., 2024). Full details of all the benchmarks and additional experimental setups are included in Appendix A.

### 3.2 DESIGN CHOICES

We conducted over 200 experiments to explore several key design choices. Table 1 summarizes our findings and provides references to the corresponding experiments that support each conclusion.

### 3.3 SHORT-RANGE BENCHMARK RESULTS

In this section, we evaluate Avey on standard autoregressive language modeling benchmarks, comparing it against Transformer++, Mamba, and RWKV-7 across three model sizes, small, medium, and large, as defined in Section 3.1. We utilize a range of widely used zero-shot downstream evaluation tasks, all detailed in Section 3.1. Table 2 summarizes the results. With small models, Avey, Mamba, and RWKV-7 outperformed Transformer++ by average margins of 1.43%, 2.41%, and

---

[5]The name is inspired from the CNN literature (Hu et al., 2018).

[6]This inequality will always hold if $m_h + m_t \geq 2d$, as is the case in our experiments (see Section 3).

Table 1: Summary of studies for key design choices and corresponding experimental references.

| Question | Answer | Experiments |
|---|---|---|
| RMSNorm or LayerNorm? | RMSNorm | Appendix H |
| LR: to decay or not to decay? | Yes, cosine decay with peak LR of $1e-3$ | Appendix I |
| Best values for sequence length $N$, split size $S$, and top-$k$ splits? | $N = 512$, $S = 64$, $k = 7$ | Appendix G |
| Activation in the enricher? | Yes, $\text{ReLU}^2$ | Appendix B |
| Activation in the contextualizer? | No | Appendix C |
| Deeper model and narrower embeddings, or shallower model and wider embeddings? | Deeper model and narrower embeddings | Appendix F |
| Weight ranked splits? | Yes, using normalized scores | Appendix K |
| Enrich embeddings before contextualization? | Yes, by 4x | Appendix K, D |
| Bypass uncontextualized features to the fuser? | Yes, 50% of each enriched embedding | Appendix K, E |
| Static or dynamic parametrization for the contextualizer? | Dynamic parametrization | Appendix K |
| Replace the contextualizer with self-attention? | No | Appendix K |

Table 2: Zero-shot performance across multiple NLP tasks.

| Model | ARC-C | ARC-E | HellaSwag | OBQA | PIQA | SIQA | Winogrande | Average |
|---|---|---|---|---|---|---|---|---|
| **Avey-153M** | **24.37** | 42.33 | 39.36 | **31.40** | 68.37 | 39.13 | 51.28 | 42.32 |
| Transformer++-152M | 23.63 | 43.17 | 39.32 | 29.80 | 67.01 | 38.89 | 50.22 | 41.72 |
| Mamba-144M | 24.17 | **43.53** | 40.55 | 30.40 | 68.32 | **39.41** | **52.72** | 42.73 |
| RWKV-7-168M | 24.17 | 43.01 | **41.55** | 29.67 | **68.72** | 39.17 | 51.09 | 42.48 |
| **Avey-496M** | 27.50 | 48.95 | 51.82 | 32.47 | 72.49 | 40.15 | 54.38 | 46.82 |
| Transformer++-488M | 26.73 | 48.09 | 52.66 | 31.73 | 72.13 | 39.93 | 55.25 | 46.65 |
| Mamba-500M | **28.64** | **51.02** | 54.15 | 34.47 | 73.03 | **40.84** | 55.49 | **48.23** |
| RWKV-7-501M | 27.13 | 49.37 | **54.54** | **36.27** | **73.58** | 39.40 | **55.72** | 48.00 |
| **Avey-1.52B** | 31.26 | 56.55 | 61.42 | 36.80 | 75.61 | 42.00 | 57.06 | 51.53 |
| Transformer++-1.5B | 30.00 | 56.29 | 63.87 | **38.00** | 76.01 | **42.24** | **61.38** | 52.54 |
| Mamba-1.4B | 32.13 | 57.74 | 63.74 | 36.85 | 76.19 | 42.00 | 60.40 | 52.72 |
| RWKV-7-1.5B | **32.94** | **59.05** | **64.43** | 37.13 | **76.84** | 41.71 | 60.06 | **53.17** |

1.82%, respectively. Mamba and RWKV-7 slightly exceeded Avey's performance, with average margins of 0.9% and 0.3%, respectively. With medium models, Avey, Mamba, and RWKV-7 again outperformed Transformer++ by averages of 0.3%, 3.4%, and 2.9%, respectively. Lastly, with large models, Avey underperformed Transformer++ by an average of 1.9%, while Mamba and RWKV-7 marginally outpaced it by 0.71% and 1.19%, respectively.

The results above assume a fixed training budget of 100B tokens. To better understand how Avey scales with increasing model size, we conducted additional experiments following the Chinchilla scaling laws (Hoffmann et al., 2022), which recommend increasing the number of training tokens proportionally with model size. Consequently, we adjusted the number of training steps and tokens to align with these laws. Appendix J outlines the configurations of the trained models, including the numbers of layers, embedding dimensions, training steps, learning rates, and total training tokens. The methodology of these experiments closely follows that of (Gu and Dao, 2023), with slight modifications (e.g., to accommodate parameter budget constraints). As demonstrated in Appendix J, Avey scales as effectively as Transformer++, particularly when both model size and token count are scaled proportionally in accordance with the Chinchilla scaling laws.

### 3.4 LONG-RANGE BENCHMARK RESULTS

We now evaluate Avey, Transformer++, Mamba, and RWKV-7 on benchmarks designed to assess performance on tasks with long-range dependencies. Specifically, we use the standard Single Needle-In-A-Haystack (S-NIAH) benchmark suite from RULER(Hsieh et al., 2024), as described in Section3.1. The S-NIAH suite includes multiple variants, notably S-NIAH-1 (pass-key retrieval) and S-NIAH-2 (number in haystack). S-NIAH-1 involves retrieving the specific value associated with

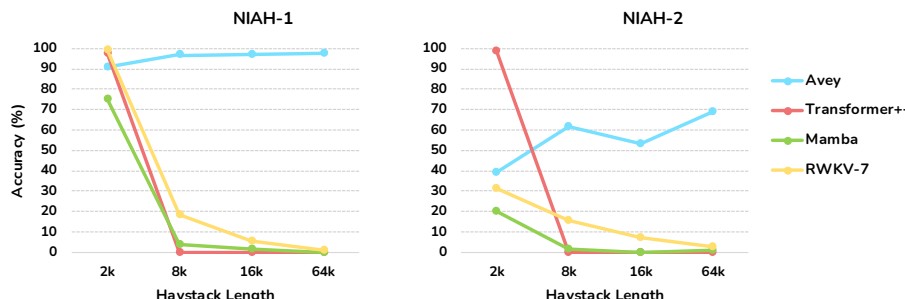

Figure 4: Performance comparison between Transformer++, Mamba, RWKV-7, and Avey on S-NIAH-1 and S-NIAH-2. The x-axis denotes the lengths of haystacks (i.e., documents with distractor texts, varying from 2k to 64k tokens). All models use 0.5B parameters. Similar results are shown in Appendix M for other model sizes.

a given key (the *pass-key*) from a distractor text (the *haystack*) containing many key-value pairs. The pass-key, serving as the *needle*, appears only once, and the model must accurately recall its corresponding value regardless of its position in the haystack. S-NIAH-2 is similar to S-NIAH-1 but poses a greater challenge, whereby the value to be retrieved is numerical (e.g., a random 9-digit number). This task requires *exact* recall, where even a single-digit error is considered incorrect, thereby testing the model's precision in extracting structured numerical information from long haystacks.

Fig. 4 demonstrates the results of Avey, Transformer++, Mamba, and RWKV-7 on both S-NIAH-1 and S-NIAH-2 benchmarks. As described in Section 3.1, Transformer++, Mamba, and RWKV-7 were all trained with a context window of 2,048 tokens. As shown, Transformer++ performs strongly on both benchmarks as long as the haystack length remains within its trained context window. Once the haystack's length exceeds its window width, Transformer++ fails to recall the correct values associated with the keys. In contrast, Mamba and RWKV-7 exhibit some ability to generalize beyond their training windows, but their performance also declines significantly as the haystack length increases far beyond those limits. On the flip side, Avey achieves good performance across both benchmarks, *despite being trained with a context window of only 512 tokens*. For instance, on S-NIAH-2 with a 64k-token haystack, Avey outperforms Mamba and RWKV-7 by averages of 85.25% and 23.6%, respectively. In addition, on S-NIAH-1 under the same 64k-token setting, Avey achieves an accuracy of 97.8%, while Mamba and RWKV-7 drop to 0% and 0.8%, respectively.

Interestingly, Avey's performance tends to improve as the haystack length increases, highlighting its strong extrapolative capability. This behavior can be attributed to the fact that as the haystack length (i.e., sequence length $N$) grows, the candidate pool from which the ranker selects the top-$k$ splits for contextualization also expands. As discussed in Appendix G, a larger $N$ enables the ranker to identify and retrieve more relevant splits while discarding less informative ones, thereby improving the overall quality of contextualization and potentially enhancing performance. This effect is further supported by the results in Appendix K, where the inclusion of the ranker led to measurable performance gains. Notably, embeddings containing a needle—whether in S-NIAH-1 or S-NIAH-2—are not processed in isolation but rather contextualized alongside other embeddings. As such, an improved quality of contextualization driven by the ranker may contribute to more accurate value recall. However, whether this mechanism fully explains Avey's increasing performance with longer haystacks remains uncertain, and further interpretability studies are needed to better understand the underlying drivers of this behavior.

## 4 RELATED WORK

Appendix O provides a comprehensive survey on related work.

## 5 CONCLUSION

In this paper, we introduced Avey, a new foundational architecture for autoregressive language modeling. Unlike traditional models, Avey relies neither on recurrence nor attention. Instead, it employs a neural approach to enrich and contextualize embeddings. Additionally, it leverages a ranker that enables the model to flexibly and effectively handle sequences of arbitrary lengths, despite being trained with only a small context window. We hope this work lays the groundwork for future research and inspires further advances in scalable and effective language modeling.

## 6 REPRODUCIBILITY

All results reported in this paper are reproducible. Section 2 specifies Avey's components in detail. The full experimental methodology is provided in Appendix A. We attach a repository with code as supplementary material. The repository includes: (1) training and evaluation scripts; (2) configuration files with the exact hyperparameters used for every experiment; (3) data preprocessing instructions and dataset references/splits; and (4) environment specifications and run scripts to regenerate all tables and figures. Using the provided commands on hardware comparable to our setup reproduces the reported numbers within expected seed variance.

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

## A EXPERIMENTAL METHODOLOGY

In this section, we describe the experimental methodology employed throughout the paper. To begin with, we adopted a *cascaded search process* to identify the best configuration of Avey. Specifically, we began with a baseline version of the neural processor—excluding the ranker—and sequentially explored several architectural design choices. After each empirical conclusion regarding a specific architectural element, we integrated that element into the processor (one at a time) and resumed the search process from the updated configuration. This cascaded process is captured chronologically in Appendices B, C, D, E, and F.

To elaborate, we started with an *expansion factor* of $4\times$ in the enricher, a *tail size* of 50% (i.e., half of each expanded embedding is forwarded to the contextualizer), RMSNorm (Zhang and Sennrich, 2019) as a normalization technique, no activation functions in the enricher and contextualizer, global batch size of 0.5M, context width of 1024, and a constant learning rate of $1e-3$. As we empirically verified and decided upon each architectural element, we updated the processor accordingly. For example, after determining that ReLU$^2$ was the most effective activation function for the enricher, we integrated it into the model and proceeded with the remaining search.

After finalizing the above exploratory set of experiments, we incorporated the ranker into the neural processor and conducted an extensive study—comprising over 138 training and inference runs—to identify the optimal sequence length (i.e., $N$), split size (i.e., $S$), and number of top-$k$ splits (i.e., $k$). The results of this sensitivity study are summarized in Appendix G. Following this, we evaluated the best normalization technique (Appendix H) as well as the optimal peak learning rate and learning rate schedule (Appendix I) for the full architecture.

All the experiments described above were conducted using a 145-million-parameter model trained on 10 billion tokens from the FineWeb dataset[7] (Hugging Face, 2023) (specifically, the sample-100BT subset of FineWeb). The results of these experiments informed the following final selection of training and model hyperparameters for Avey across three parameter scales, 153M (*small*), 496M (*medium*), and 1.52B (*large*).

**Training and Model Hyperparameters of AVEY:**

- **Training hyperparameters:**
  - Optimizer: AdamW
  - Betas: (0.9, 0.95)
  - Epsilon: $1e-12$
  - Peak learning rate: $1e-3$
  - Schedule: Cosine decay to 10% of the peak learning rate, with no warmup
  - Batch size: 0.5M for the small and medium models, and 1M for the large model
  - Gradient norm clip: 1.0
  - Weight decay: 0.1 (applied only to matrices)
- **Model hyperparameters:**
  - **All models:**
    - Context width: 512
    - Split size ($S$): 64
    - Number of top-$k$ splits: 7
    - Vocabulary size: 50,304

---

[7]This dataset is released under the Open Data Commons Attribution License (ODC-By) v1.0.

- Expansion factor: 4
- Tail size: 0.5
- **Small model (153M parameters):**
  - Embedding dimension: 768
  - Number of layers: 26
- **Medium model (496M parameters):**
  - Embedding dimension: 768
  - Number of layers: 104
- **Large model (1.52B parameters):**
  - Embedding dimension: 2048
  - Number of layers: 48

For baselines, we compared Avey against three leading open-source models, namely, Mamba (Implementation, 2023a), RWKV-7 (Implementation, 2023b), and Transformer++ (Karpathy, 2023). For Transformer++, we implemented the strongest architectural recipe known to us, incorporating rotary positional encodings (Su et al., 2024), SwiGLU MLPs (Shazeer, 2020), and RMSNorm in place of LayerNorm (Zhang and Sennrich, 2019). All models were trained using their best-known hyperparameters (see details below, assuming three model sizes, *small*, *medium*, and *large*) under a fixed budget of 100 billion tokens drawn from the aforementioned FineWeb dataset [8]. For consistency and comparability, we used the p50k_base tokenizer (OpenAI, 2022) across all the models, as it aligns with the GPT-2-derived token counts reported for this dataset.

**Training and Model Hyperparameters of TRANSFORMER++:**

- **Training hyperparameters:**
  - Optimizer: AdamW
  - Betas: (0.9, 0.95)
  - Epsilon: $1e{-}12$
  - Peak learning rates:
    - Small model: $3e{-}3$
    - Medium model: $1.5e{-}3$
    - Large model: $1.25e{-}3$
  - Schedule: A linear warmup for 10% of steps, followed by cosine decay to 10% of the peak learning rate
  - Batch size: 0.5M for the small and medium models, and 1M for the large model
  - Gradient norm clip: 1.0
  - Weight decay: 0.1 (only applied to matrices)
- **Model hyperparameters:**
  - **All models:**
    - Context width: 2048
    - Vocabulary size: 50,304
    - Intermediate size in FFN: $4\times$ the embedding dimension
  - **Small model (152M parameters):**
    - Embedding dimension: 768
    - Number of layers: 12
    - Number of heads: 12
  - **Medium model (488M parameters):**
    - Embedding dimension: 1024
    - Number of layers: 26
    - Number of heads: 16
  - **Large model (1.5B parameters):**

---

[8]More precisely, all models, including Avey, Transformer++, Mamba, and RWKV-7 were trained for 1 epoch over the sample-100BT subset of FineWeb.

- Embedding dimension: 1664
- Number of layers: 32
- Number of heads: 16

**Training and Model Hyperparameters of MAMBA:**

- **Training hyperparameters:**
  - Optimizer: AdamW
  - Betas: (0.9, 0.95)
  - Epsilon: $1e{-}12$
  - Peak learning rates:
    - Small model: $3e{-}3$
    - Medium model: $1.5e{-}3$
    - Large model: $1.0e{-}3$
  - Schedule: A linear warmup for 10% of steps, followed by cosine decay to 10% of the peak learning rate
  - Batch size: 0.5M for the small and medium models, and 1M for the large model
  - Gradient norm clip: 1.0
  - Weight decay: 0.1 (applied only to matrices)
- **Model hyperparameters:**
  - **All models:**
    - Context width: 2048
    - Vocabulary size: 50,304
    - All hyperparameters other than the ones specified are left at their default values according to (Implementation, 2023a)
  - **Small model (144M parameters):**
    - Embedding dimension: 768
    - Number of layers: 28
  - **Medium model (500M parameters):**
    - Embedding dimension: 1280
    - Number of layers: 42
  - **Large model (1.4B parameters):**
    - Embedding dimension: 2048
    - Number of layers: 52

**Training and Model Hyperparameters of RWKV-7:**

- **Training hyperparameters:**
  - Optimizer: AdamW
  - Betas: (0.9, 0.95)
  - Epsilon: $1e{-}12$
  - Peak learning rates:
    - Small model: $6e{-}4$
    - Medium model: $4e{-}4$
    - Large model: $4e{-}4$
  - Schedule: Cosine decay to 10% of the peak learning rate, with no warmup
  - Batch size: 1M for the small and medium models, and 2M for the large model
  - Gradient norm clip: 1.0
  - Weight decay: 0.1 (applied only to matrices)
- **Model hyperparameters:**
  - **All models:**
    - Context width: 2048

Table 3: Avey's performance *without* and *with* an activation function in the **enricher**. The study involves only the neural processor within Avey and trains it on 10B tokens.

| Configuration | Perplexity | ARC-C | ARC-E | HellaSwag | OBQA | PIQA | SIQA | Winogrande | Average |
|---|---|---|---|---|---|---|---|---|---|
| No Activation | 37.65 | 22.53 | 35.19 | 28.95 | 27.40 | 60.45 | 36.13 | 48.70 | 37.05 |
| GELU | **30.10** | 23.04 | 37.88 | 31.37 | 26.00 | 63.55 | 36.95 | 51.85 | 38.66 |
| ReLU | 31.02 | 22.87 | 37.21 | 31.35 | 28.00 | 62.79 | 38.13 | 48.86 | 38.46 |
| ReLU$^2$ | 30.18 | 24.15 | 38.76 | 32.08 | 28.00 | 63.76 | 38.28 | 50.12 | **39.31** |
| SiLU | 30.81 | 22.18 | 38.43 | 31.30 | 28.20 | 62.30 | 37.05 | 53.20 | 38.95 |

- Vocabulary size: 50,304
- All hyperparameters other than the ones specified are left at their default values according to (Implementation, 2023b)
- **Small model (168M parameters):**
  - Embedding dimension: 768
  - Number of layers: 12
- **Medium model (501M parameters):**
  - Embedding dimension: 1024
  - Number of layers: 30
- **Large model (1.5B parameters):**
  - Embedding dimension: 2048
  - Number of layers: 24

To compare all models, we employed a suite of widely used NLP benchmarks, including ARC-E and ARC-C (for scientific reasoning and reading comprehension) (Clark et al., 2018), HellaSwag (for commonsense inference) (Zellers et al., 2019), PIQA (for physical reasoning) (Bisk et al., 2020), OBQA (for open-book science reasoning) (Mihaylov et al., 2018), SIQA (for social interaction understanding) (Sap et al., 2019), and Winogrande (for coreference and commonsense reasoning) (Sakaguchi et al., 2021). In addition, we evaluated long-context retrieval capabilities using the standard Single Needle-In-A-Haystack (S-NIAH) benchmark suite from RULER (Hsieh et al., 2024), which measures a model's ability to extract pass-keys from large distractor corpora, with sequence lengths ranging from 2k to 64k tokens. All evaluations were conducted using the widely adopted LM Evaluation Harness from EleutherAI (Gao et al., 2021), consistent with the prior work in the field.

For all models, we reported performance in terms of benchmark accuracy[9]. Specifically, we used normalized accuracy (acc-norm) from the LM Evaluation Harness whenever available. For each model, the reported score on each NLP benchmark is the average accuracy across its final three checkpoints (taken at 90B, 95B, and 100B tokens) to account for variability due to training randomness. Complete benchmark results across these checkpoints, along with key summary statistics and discussions, are provided in Appendix L.

Finally, all training and evaluation runs were executed on 208 NVIDIA H200 GPUs, with mixed-precision (bfloat16) enabled for training. The total training time for all models—Avey, Transformer++, Mamba, and RWKV-7—across the three presented model sizes and 100B training tokens is estimated to be approximately 80–90 hours, assuming optimal parallelization across the 208 GPUs and using the implementations referenced above. To avoid potential sources of randomness and ensure consistency across results, we disabled Torch Compile during all design choice and sensitivity experiments. For the ablation studies and final training runs, however, Torch Compile was enabled whenever possible to accelerate training. Additionally, we fixed the random seed to 11 (arbitrarily chosen) for all training runs to further reduce variability due to stochastic effects.

## B  ACTIVATION OR NO ACTIVATION IN THE ENRICHER

In this study, we illustrate the performance of Avey *with* and *without* an activation function in the enricher. To this end, we trained a model with 145 million parameters using 10 billion tokens from

---

[9]Additionally, throughout the paper, all reported perplexity values specifically refer to training perplexity.

Table 4: Avey's performance *without* and *with* an activation function in the **contextualizer**. The study involves only the neural processor within Avey, trains it on 10B tokens, and uses $ReLU^2$ within the enricher, capitalizing on the results shown in Table 3.

| Configuration | Perplexity | ARC-C | ARC-E | HellaSwag | OBQA | PIQA | SIQA | Winogrande | Average |
|---|---|---|---|---|---|---|---|---|---|
| No Activation | **30.18** | 24.15 | 38.76 | 32.08 | 28.00 | 63.76 | 38.28 | 50.12 | **39.31** |
| GELU | 30.64 | 22.01 | 37.54 | 31.23 | 27.20 | 64.74 | 37.21 | 50.67 | 38.66 |
| ReLU | 30.30 | 23.29 | 37.84 | 31.82 | 26.60 | 64.15 | 38.13 | 50.59 | 38.92 |
| $ReLU^2$ | 31.05 | 22.35 | 38.80 | 30.78 | 27.20 | 63.49 | 37.31 | 52.01 | 38.85 |
| SiLU | 30.92 | 23.63 | 36.74 | 31.51 | 27.40 | 64.20 | 36.18 | 50.04 | 38.53 |

Table 5: The effect of the expansion factor on Avey's performance. The study involves only the neural processor within Avey, trains it on 10B tokens, adopts $ReLU^2$ within the enricher, and does not use an activation function within the contextualizer, building upon the results portrayed in Tables 3 and 4.

| Expansion | Perplexity | ARC-C | ARC-E | HellaSwag | OBQA | PIQA | SIQA | Winogrande | Average |
|---|---|---|---|---|---|---|---|---|---|
| 2× | 30.85 | 23.29 | 36.83 | 30.92 | 26.20 | 63.98 | 37.72 | 51.78 | 38.67 |
| 4× | 30.18 | 24.15 | 38.76 | 32.08 | 28.00 | 63.76 | 38.28 | 50.12 | **39.31** |
| 8× | **30.00** | 23.21 | 37.79 | 31.48 | 26.20 | 63.82 | 36.59 | 50.75 | 38.55 |

the Fineweb dataset (Hugging Face, 2023). The model employs an *expansion factor* of $4\times$ in the enricher (i.e., each embedding dimension is expanded fourfold by the enricher), a *tail size* of 50% (i.e., half of each expanded embedding is forwarded to the contextualizer), RMSNorm (Zhang and Sennrich, 2019) as a normalization technique, and no activation function in the contextualizer. Additionally, the context width (i.e., the maximum number of tokens that can be input to and processed by the contextualizer simultaneously) is set to 1024 and a constant learning rate of $1e-3$ is maintained throughout training.

The study excludes the ranker and focuses solely on the neural processor. Besides, it evaluates four activation functions, namely, GELU (Hendrycks and Gimpel, 2016), ReLU (Nair and Hinton, 2010), $ReLU^2$ (Chowdhery et al., 2022), and SiLU (Ramachandran et al., 2017). All other experimental settings follow the methodology detailed in Appendix A. Table 3 summarizes the results. As shown, $ReLU^2$ yielded an improvement in performance versus a baseline with no activation, hence, was adopted as the default activation function for the enricher throughout our experiments presented in Sections 3.3 and 3.4. It is important to note, however, that the lowest perplexity was provided by GELU and not $ReLU^2$ (although the difference in perplexity was very minimal). While perplexity quantifies how effectively the model predicts the next token in the training dataset, it remains a proxy for overall modeling capability and does not always *precisely* predict downstream task performance.

## C   ACTIVATION OR NO ACTIVATION IN THE CONTEXTUALIZER

We now evaluate Avey *with* and *without* an activation function in the contextualizer. We use the same experimental settings outlined in Appendix B and add to that $ReLU^2$ as an activation function in the enricher, capitalizing on the findings therein. We also experiment with four activation functions, namely, GELU, ReLU, $ReLU^2$, and SiLU. Results are summarized in Table 4. As illustrated, the best performance was achieved without any activation function in the contextualizer, thus was employed as the default configuration in all our experiments reported in Sections 3.3 and 3.4.

## D   WHAT IS THE BEST EXPANSION FACTOR?

In this study, we vary the *expansion factor* in the enricher, defined as the degree to which each input embedding is expanded. Specifically, we evaluate several expansion factors, ranging from $2x$ to $8x$ as shown in Table 5, while keeping the total model parameter count constant (e.g., with a $2x$ expansion factor we use 34 layers, while with a $4x$ one we utilize 20 layers). The experimental setup follows the methodology outlined in Appendix B, but incorporates $ReLU^2$ as an activation

Table 6: The effect of the tail size on Avey's performance. The study involves only the neural processor within Avey, trains it on 10B tokens, adopts ReLU$^2$ within the enricher, does not use an activation function within the contextualizer, and utilizes an expansion factor of $4x$, as recommended by the findings demonstrated in Tables 3, 4, and 5.

| Tail Size | Perplexity | ARC-C | ARC-E | HellaSwag | OBQA | PIQA | SIQA | Winogrande | Average |
|---|---|---|---|---|---|---|---|---|---|
| 10% | 34.23 | 21.59 | 36.78 | 30.34 | 27.0 | 63.60 | 37.15 | 50.28 | 38.11 |
| 30% | 30.91 | 23.55 | 37.33 | 31.41 | 29.4 | 64.25 | 37.15 | 51.22 | 39.19 |
| 50% | 30.18 | 24.15 | 38.76 | 32.08 | 28.0 | 63.76 | 38.28 | 50.12 | **39.31** |
| 70% | **29.79** | 23.04 | 38.26 | 31.68 | 28.4 | 63.55 | 37.36 | 49.96 | 38.89 |
| 90% | 30.20 | 23.04 | 38.38 | 32.13 | 27.6 | 64.04 | 37.67 | 50.91 | 39.11 |

Table 7: Avey's performance across different model configurations, including wider embedding dimensions (e.g., 1536 under 0.5B-parameter model) with shallower layers (e.g., 24 layers under 0.5B-parameter model), or narrower embedding dimensions (e.g., 768 under 0.5B-parameter model) with deeper layers (e.g., 90 layers under 0.5B-paramter model). The models with 140M, 0.5B, and 1.5B parameters are referred to as *small Avey*, *medium Avey*, and *large Avey* in the text.

| # Params | Embed. | # Layers | Perplexity | ARC-C | ARC-E | HellaSwag | OBQA | PIQA | SIQA | Winogrande | Average |
|---|---|---|---|---|---|---|---|---|---|---|---|
| | 512 | 40 | 31.14 | 22.53 | 37.58 | 31.05 | 28.6 | 63.87 | 37.46 | 52.64 | 39.10 |
| 140 M | 768 | 20 | **30.18** | 24.15 | 38.76 | 32.08 | 28.0 | 63.76 | 38.28 | 50.12 | **39.31** |
| | 1024 | 11 | 31.46 | 23.46 | 38.30 | 30.78 | 27.0 | 63.93 | 37.72 | 48.86 | 38.58 |
| | 768 | 90 | **23.02** | 23.55 | 42.05 | 38.61 | 30.2 | 66.10 | 39.20 | 51.78 | 41.64 |
| 0.5 B | 1024 | 54 | 23.27 | 24.40 | 41.92 | 38.46 | 29.2 | 67.19 | 38.74 | 51.46 | 41.62 |
| | 1536 | 24 | 23.51 | 23.98 | 42.85 | 37.79 | 29.4 | 66.97 | 38.89 | 51.78 | **41.67** |
| | 1536 | 80 | 19.97 | 25.26 | 45.50 | 44.56 | 30.2 | 68.61 | 40.02 | 52.33 | 43.78 |
| 1.5 B | 2048 | 48 | **19.84** | 25.77 | 46.55 | 44.99 | 31.6 | 69.42 | 40.17 | 52.17 | **44.38** |
| | 2560 | 30 | 20.23 | 26.62 | 45.16 | 43.91 | 29.2 | 69.10 | 39.82 | 52.09 | 43.70 |

function in the enricher and omits any activation function in the contextualizer, aligning with the findings reported in Appendices B and C. As depicted in the table, an expansion factor of $4x$ yielded the best performance, hence, was set as the default configuration in the enricher throughout all our experiments presented in Sections 3.3 and 3.4.

# E  WHAT IS THE BEST TAIL SIZE?

We now examine the impact of forwarding a tail portion of each enriched embedding to the contextualizer. More precisely, we vary the size of this tail portion, referred to as the *tail size*, from 10% to 90% of each enriched embedding, as illustrated in Table 6. The study follows the experimental setup described in Appendix B and employs ReLU$^2$ as an activation function in the enricher, no activation function in the contextualizer, and an expansion factor of $4x$, based on the findings presented in Appendices B, C, and D, respectively. As depicted in Table 6, the best performance was accomplished using a tail size of 50%, thus was adopted as the default configuration for Avey in all our experiments reported in Sections 3.3 and 3.4.

# F  DEEPER MODELS AND NARROWER EMBEDDINGS, OR SHALLOWER MODELS AND WIDER EMBEDDINGS

The objective of this study is to determine whether a narrower embedding dimension with a greater model depth yields better or worse performance than a wider embedding dimension with fewer layers. The study utilizes the experimental setup described in Appendix B and leverages the findings presented in Appendices B, C, D, and E. Consequently, it utilizes ReLU$^2$ as an activation function in the enricher, no activation function in the contextualizer, an expansion factor of $4x$, and a tail size of 50%.

To begin with, we evaluated a small Avey model (referred to as *small Avey*) with 140 million parameters, using three different embedding dimensions, 512, 768, and 1024. These configurations resulted

in 40, 20, and 11 layers, respectively, to maintain a constant parameter count. Table 7 summarizes the results. As illustrated, the configuration with an embedding dimension of 768 and a layer count of 20 outperformed the other two configurations.

Afterwards, we assessed a larger Avey model with 500 million parameters (referred to as *medium Avey*), using three different embedding dimensions, 768, 1024, and 1536. These configurations resulted in 90, 54, and 24 layers, respectively, while keeping the total parameter count constant. As shown in Table 7, the setup with an embedding dimension of 768 and 90 layers yielded the best performance among the three tested ones.

Finally, we examined an even larger Avey model with 1.5 billion parameters (referred to as *large Avey*), using three different embedding dimensions, 1536, 2048, and 2560. To maintain a constant parameter count across configurations, these dimensions corresponded to 80, 48, and 30 layers, respectively. As portrayed in Table 7, the configuration with an embedding dimension of 2048 and 48 layers delivered the best performance among the three considered configurations.

The above results suggest a trend, whereby wider embedding dimensions (e.g., 1024 in small Avey; 1563 in medium Avey; and 2560 in large Avey) paired with shallower architectures (e.g., 11 layers in small Avey; 24 layers in medium Avey; and 30 layers in large Avey) tend to underperform deeper models (e.g., 40 and 20 layers in small Avey; 90 and 54 layers in medium Avey; and 80 and 48 layers in large Avey) with narrower embeddings (e.g., 512 and 768 in small Avey; 768 and 1024 in medium Avey; and 1536 and 2048 in large Avey). As such, in all our experiments discussed in Sections 3.3 and 3.4, we employed deeper models with narrower embedding dimensions, namely, the best performing small Avey, medium Avey, and large Avey in Table 7.

Interestingly, Table 7 also highlights that certain benchmarks benefit more from increased model capacity than others. For instance, a commonsense reasoning benchmark like HellaSwag demonstrates performance improvements of 20.36% and 28.5% under 0.5B-parameter and 1.5B-parameter models, respectively, compared to a 140M-parameter baseline. In contrast, a question-answering benchmark such as SIQA exhibits only a modest gain of 2.4% under *both* 0.5B-parameter and 1.5B-parameter models relative to the 140M-parameter baseline, suggesting less sensitivity to model size.

## G    WHAT ARE THE BEST SEQUENCE LENGTH, SPLIT SIZE, AND TOP-$k$ VALUES?

We now analyze how Avey's perplexity and overall performance are affected by variations in three key parameters, the ranker's top-$k$ selected splits, the split size $S$, and the sequence length $N$. We consider three values for $N$, 256, 512, and 1024. For each $N$, the split size $S$ is grown geometrically, starting from 16 and doubling at each step, up to the maximum permissible value $N/2$. Subsequently, for any $N$ and $S$, the number of top-ranked splits $k$ can range from 1 (i.e., contextualizing the current split with one additional relevant split) up to $N/S - 1$. To tame the quantity of experiments, we increase $k$ arithmetically from 1 to a maximum of 15, whenever possible, using an increment of 2. Finally, we use the experimental configurations recommended in Appendices B, C, D, E, and F.

As shown in Table 8, Avey's perplexity is highest when both $S$ and $k$ are very small (e.g., $S = 16$ and $k = 1$). While a small $S$ can help filter out irrelevant embeddings and *denoise* contextualization, pairing it with a very small $k$ can deprive the contextualizer of sufficient context to build expressive representations[10]. To expand the context (i.e., increase its width $S(k + 1)$) and enrich the resulting embeddings, either $S$ or $k$ can be increased. For example, as we increased $k$ under $N = 256$ and $S = 16$, perplexity decreased and benchmark performance improved. However, a larger context does not always yield better outcomes, especially when involving a high proportion of irrelevant embeddings. This behavior was observed when $k$ was increased beyond 3 under $N = 1024$ and $S = 128$, resulting in higher perplexity and diminished downstream accuracy.

In contrast to $S$ and $k$, the sequence length $N$ determines the size of the candidate pool from which the ranker selects the top-$k$ splits for any current split during training. A larger $N$ allows the ranker to reach farther back in the sequence history, potentially retrieving more relevant splits and lowering perplexity. For example, increasing $N$ from 512 to 1024, while holding $S = 16$ and $k = 15$

---

[10]Recall from Section 2.2.2 that the context width $C$ is defined as $C = S(k + 1)$.

constant, reduced perplexity by $5.4\%$. Nevertheless, lower perplexity does not always translate into improved downstream performance. For instance, with $N = 512$ and $S = 16$, $k = 15$ yielded the lowest perplexity, yet $k = 5$ achieved higher benchmark accuracy. As noted in Appendix B, the loss remains only a proxy for overall modeling capability and does not always exactly predict downstream task performance. As demonstrated in Table 8, the best empirical performance was obtained with $N = 512$, $S = 64$, and $k = 7$, which was, accordingly, adopted as Avey's default configuration.

## H    RMSNORM OR LAYERNORM

In all previous runs across the appendices, we used RMSNorm (Zhang and Sennrich, 2019) as a normalization technique. We now test Avey with another standard normalization method, namely, LayerNorm (Ba et al., 2016). To begin with, we note that Avey normalizes input embeddings before each layer (as illustrated in Fig. 3) and output embeddings *once* after the final layer and prior to token prediction.

We evaluate each normalization technique using Avey's complete architecture, including its neural processor and ranker. The ranker is configured according to the best-performing setting identified in Appendix G (i.e., sequence length $N = 512$, split size $S = 64$, and number of top-ranked splits $k = 7$). The neural processor employs the optimal configurations reported in Appendices B, C, D, E, and F. As before, we trained the resulting model with 153 million parameters on 10 billion tokens from the Fineweb dataset. Table 9 summarizes the results. As shown, RMSNorm slightly outperforms LayerNorm on average, and is therefore adopted in Avey and used consistently across all the experiments presented in Sections 3.3 and 3.4.

## I    LEARNING RATE: TO DECAY OR NOT TO DECAY?

In all previous runs across the appendices, we used a constant learning rate of $1e{-}3$. In this study, we evaluate the performance of Avey under varying maximum learning rates and learning rate schedules. Specifically, we compare two schedules, cosine decay and constant learning rate (which can be viewed as cosine decay with an infinite cycle length, effectively eliminating any decay). The study adopts the experimental configurations suggested in Appendices B, C, D, E, and F. In addition, it employs Avey's complete architecture, including its neural processor and ranker, with the best configuration from Table 8.

As shown in Table 10, cosine decay consistently achieves lower losses than the constant schedule across the tested learning rates. This observation aligns with findings from the *Chinchilla* paper (Hoffmann et al., 2022), which indicates that when the cosine cycle length significantly exceeds the total number of training steps (by at least 25%), model performance tends to deteriorate. Notably, the longest cycle length arises when the schedule is constant. In contrast, setting it to approximately match the training duration yields the best final loss (Hoffmann et al., 2022).

To this end, we adopt a cosine decay schedule with a peak learning rate of $1e{-}3$ for Avey, especially that it provides the lowest loss across all the runs. We note, however, that Table 10 also shows that the lowest loss does not correspond to the best downstream task performance. For instance, the loss of 3.308 under the constant learning rate schedule resulted in a slightly better benchmark performance than the lower loss of 3.218 under the cosine decay schedule, both using the same peak learning rate. While constant learning rates can be effective for short or exploratory runs (this study uses only 10 billion tokens), it is generally the case that, as the number of training tokens increases, the learning rate must decrease to allow the optimizer to *settle* into a lower-loss region (You et al., 2019). Hence, schedules with decay are typically favored for longer or large-scale training runs (Hoffmann et al., 2022; Bergsma et al., 2025).

## J    SCALING LAWS

In this section, we present a scaling law study comparing how well Avey, Transformer++, Mamba, and RWKV-7 scale with increasing compute. For Avey, we use the full architecture, including both the ranker and neural processor. All models are trained at three different sizes, as defined in

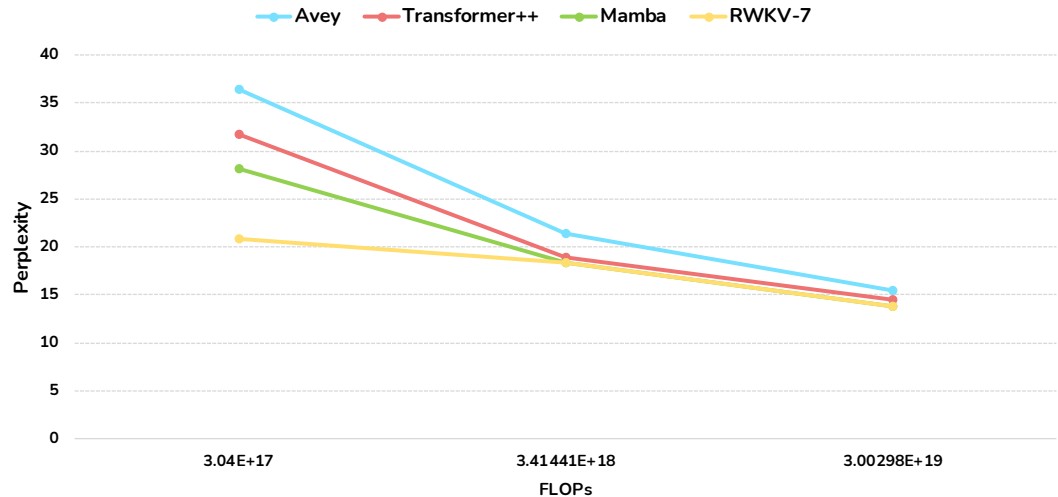

Figure 5: Scaling law results, comparing how perplexity decreases as compute increases, assuming three model sizes of 150M, 500M, and 1.5B parameters and a proportional increase in the number of training tokens with model size, following the Chinchilla scaling laws.

Appendix A and summarized in Table 11. To ensure compute-optimal scaling, we proportionally increase the number of training tokens with model size, following the Chinchilla scaling laws (Hoffmann et al., 2022). Specifically, and in line with the methodology from (Gu and Dao, 2023), we use 2B, 7B, and 20B tokens to train models with approximately 150M, 500M, and 1.5B parameters, respectively. Lastly, we employ the same batch size across all the models to control for variability in the number of gradient update steps, especially because of training with a limited number of tokens.

Fig. 5 presents the scaling results. The x-axis represents the total training compute budget, calculated as the product of the number of training tokens and model parameters, which serves as a proxy for the total FLOPs required to train each model. As shown, Avey exhibits the steepest decline in perplexity as compute increases. Although it begins with a relatively high perplexity[11], it improves more rapidly than the other models, indicating strong scaling behavior and greater benefit from additional compute. Following Avey, Transformer++ demonstrates the next-best scaling trend, outpacing Mamba and RWKV-7. While Mamba achieves relatively low perplexity at smaller compute budgets, it does not scale as effectively as Avey or Transformer++. Finally, RWKV-7 performs well at low compute but shows the flattest scaling curve, suggesting it gains the least from additional training compute.

## K   ABLATION STUDY

In this study, we perform a series of ablation experiments on the core components of Avey, leveraging the best configurations identified in Appendices B, C, D, E, F, H, and I. All experiments are conducted using Avey's complete architecture, comprising both the ranker and neural processor, with the small model variant (153 million parameters) as the baseline. For this study, we trained this model on 10 billion tokens from the FineWeb dataset, using the training methodology and hyperparameters detailed in Appendix A.

We begin by examining the effect of dynamic parameterization on both perplexity and downstream benchmark performance. As described in Section 2.2.2, dynamic parameterization allows each neuron in the contextualizer network to compute a cosine similarity between its input embedding and

---

[11] Avey can achieve substantially lower perplexity under alternative configurations. For example, the small model (150M parameters) with sequence length $N = 1024$, split size $S = 16$, and top-$k = 15$ achieves much lower perplexity as shown in Table 8. The configuration used in this experiment– and in Sections 3.3 and 3.4, was selected based on its strong downstream benchmark performance, rather than optimal perplexity. As discussed in Appendix B, perplexity serves as a useful proxy for modeling capability, but does not always align perfectly with downstream task accuracy.

the embeddings of all other neurons, weight those embeddings accordingly, and aggregate them via a learned weighted sum. This mechanism induces selectivity, as defined in (Gu and Dao, 2023), into the neural processor, thereby making its parameterization dynamic (or input-dependent). Table 12 summarizes the results. Disabling this component results in a 14.3% increase in perplexity and a 0.8% drop in average performance, highlighting its importance.

Second, we evaluate the impact of the partial-embedding bypassing technique introduced in Section 2.2.1. This method involves forwarding a portion of each expanded embedding directly to the fuser, allowing raw, distinctive features to be preserved and potentially serve in promoting more diverse representations. As shown in Table 12, removing this mechanism results in an 8.5% increase in perplexity and a 2.2% drop in average performance, underscoring its significance.

Third, we set the expansion factor in the enricher to 1, effectively disabling the expansion of input embeddings. As illustrated in Table 12, this modification results in a 33.1% increase in perplexity and a 5.2% drop in average performance, corroborating the critical role of embedding expansion in the model's effectiveness.

Fourth, we remove the weighting of each selected split by its corresponding normalized MaxSim score, thereby preventing the contextualizer from scaling each split's contribution during contextualization. As depicted in Table 12, this adjustment leads to a 3.8% increase in perplexity and a 1.37% drop in average performance, indicating the importance of this technique.

Fifth, we evaluate Avey *without* the ranker to assess its impact on downstream task performance, beyond its primary role of enabling effective extrapolation past the trained context window. As shown in Table 12, the ranker does indeed enhance the neural processor's performance, primarily by improving the quality of contextualization through more meaningful cross-token interactions. We note, however, the slight increase in loss (by 0.5%), which again highlights (as in Appendix B) the discrepancy between the objectives of pertaining and downstream tasks.

Finally, we replace Avey's neural processor with self-attention to assess the relative contribution of each component to Avey's overall performance, given that both are designed to pursue cross-token interactions. As illustrated in Table 12, this substitution leads to a 4.6% increase in perplexity and a 2.1% decline in average performance, underscoring the significance of the neural processor and suggesting that self-attention is less effective within Avey's architectural framework.

## L    ADDITIONAL SHORT-RANGE BENCHMARK RESULTS

To mitigate the effects of fluctuations in pre-training loss and downstream benchmark scores, we reported in Section 3.3 average results across the final three checkpoints—taken at 5 billion token intervals (i.e., at 90B, 95B, and 100B tokens)—for all models evaluated, namely, Avey, Transformer++, Mamba, and RWKV-7. In this section, we provide the detailed performance scores for each model at each checkpoint in Table 13. In addition, we summarize the mean, standard deviation, standard error, and 95% confidence interval for each model, computed across the three checkpoints, in Table 14. The illustrated statistical results reveal meaningful variance between models and across runs of the same model. For instance, while Mamba achieves the highest mean score of 42.73 among all the models in the small parameter regime ($\sim$150M parameters), it also exhibits a relatively wide confidence interval (42.06, 43.40) and a moderate standard deviation, highlighting nontrivial variability in performance across checkpoints.

Regarding variability across models, Table 14 shows overlapping confidence intervals, indicating that model rankings—particularly which model achieves the highest mean performance—could shift under minor experimental changes (e.g., random initialization, stochastic optimization, etc.). For example, in the small model regime, while Avey does not surpass Mamba in mean performance, their confidence intervals substantially overlap in the range (42.06, 43.24), suggesting that the two models are statistically comparable and that Avey could outperform Mamba in some runs. Similarly, a narrow but meaningful overlap exists between Avey and RWKV-7 in the range (42.46, 42.51), implying that Avey may occasionally match or slightly exceed RWKV-7 in certain cases. Lastly, although Mamba has the highest mean in this setting, its confidence interval also overlaps with RWKV-7, indicating that the difference in performance between the two models is not statistically significant and that RWKV-7 could match or slightly outperform Mamba in some runs.

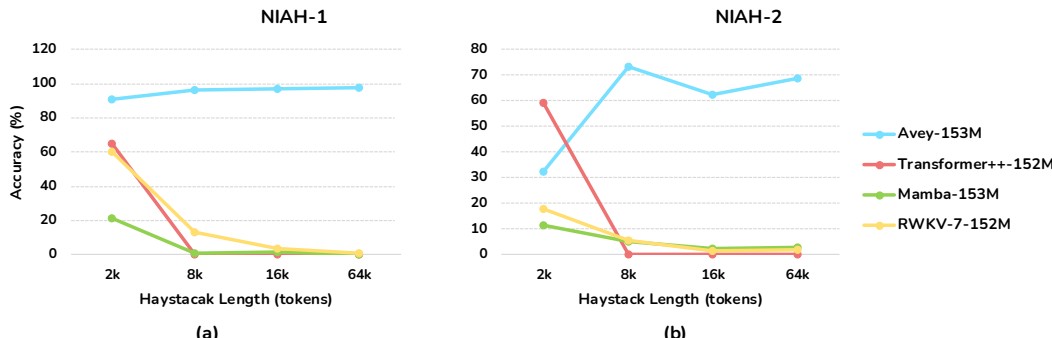

Figure 6: Performance comparison between Transformer++, Mamba, RWKV-7, and Avey on S-NIAH-1 and S-NIAH-2. The x-axis denotes the lengths of haystacks (i.e., documents with distractor texts, varying from 2k to 64k tokens). All models use ∼**150M parameters**.

In the medium model regime (∼500M parameters), Avey outperforms Transformer++ in mean performance, but they are statistically comparable. In contrast, the performance gap between Avey and both Mamba and RWKV-7 is statistically significant at the 95% confidence level, indicating that both models clearly outperform Avey in this setting. In the large model regime (∼1.5B parameters), while Avey does not outpace Transformer++ in average performance, their confidence intervals overlap substantially, suggesting that Avey could potentially surpass Transformer++ in some runs. There is also a limited overlap between Avey and Mamba, indicating that while Mamba generally performs better, Avey might outperform it in certai n cases. In contrast, the difference between Avey and RWKV-7 is statistically significant at the 95% level, confirming that RWKV-7 consistently outperforms Avey in this setting. Finally, although RWKV-7 has a slightly higher mean than Mamba (53.17 vs. 53.12), the meaningful overlap in their confidence intervals implies that the difference between them is not statistically significant, and either model could outperform the other depending on minor experimental factors.

## M  ADDITIONAL LONG-RANGE BENCHMARK RESULTS

In Section 3.4, we presented results for Avey, Transformer++, Mamba, and RWKV-7 under the medium parameter regime (∼500M parameters) on the standard Single Needle-In-A-Haystack (S-NIAH) benchmark suite from RULER (Hsieh et al., 2024), which is designed to evaluate models' abilities to handle long-range dependencies. The S-NIAH benchmark, along with two of its common variants—S-NIAH-1 and S-NIAH-2—was described in detail in Section 3.4. In this section, we extend our analysis by reporting results under two additional model regimes, small (∼150M parameters) in Fig. 6 and large (∼1.5B parameters) in Fig. 7. Akin to the experiment in Section 3.4, Transformer++, Mamba, and RWKV-7 were trained with a context window of 2048 tokens, while Avey was trained with a shorter window of only 512 tokens.

In both the small and large model regimes, under S-NIAH-1 and S-NIAH-2, Transformer++, Mamba, and RWKV-7 perform well when the haystack length is 2k, fitting within their trained context windows. Yet, Mamba consistently underperforms Transformer++ and RWKV-7, likely due to solely relying on recurrence, which somehow treats the entire input uniformly, making the model more susceptible to distractions from irrelevant tokens. In contrast, RWKV-7, which combines recurrence with attention, performs better than Mamba but remains inferior to Transformer++, potentially because the attention mechanism allows it to prioritize tokens relevant to the needle, while the recurrent component may still contribute to signal dilution. Transformer++, relying exclusively on full attention, achieves the best performance within the context window by effectively focusing on relevant tokens without interference from recurrence-based mechanisms. Nonetheless, once the haystack length exceeds the models' context windows, all the three models exhibit a substantial drop in performance. Mamba and RWKV-7, however, show minimal generalization beyond their training limits compared to Transformer++, as previously discussed in Section 3.4.

Compared to Transformer++, Mamba, and RWKV-7, Avey generalizes far beyond its trained context window on both S-NIAH-1 and S-NIAH-2 across all parameter regimes, underscoring its strong

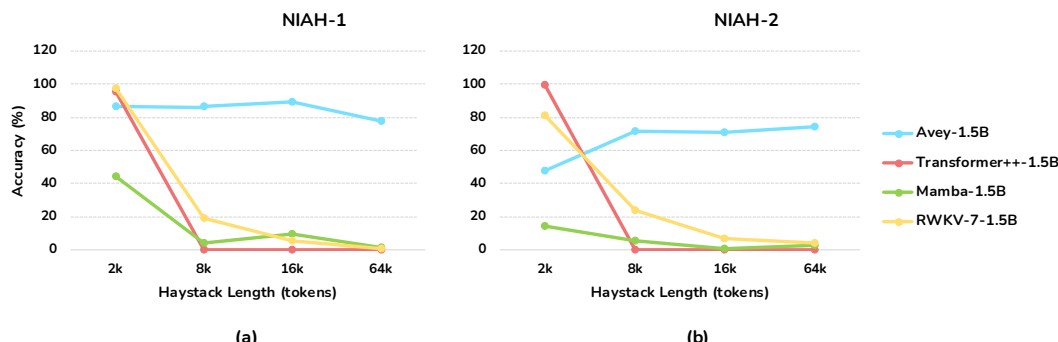

Figure 7: Performance comparison between Transformer++, Mamba, RWKV-7, and Avey on S-NIAH-1 and S-NIAH-2. The x-axis denotes the lengths of haystacks (i.e., documents with distractor texts, varying from 2k to 64k tokens). All models use ~**1.5B parameters**.

extrapolative capabilities (as also shown in Section 3.4). Notably, this holds despite Avey being trained with a context window of only 512 tokens. For example, in the small parameter regime, Avey achieves an accuracy of 97.8% on S-NIAH-1 with a 64k-token haystack, while Transformer++, Mamba, and RWKV-7 drop to 0%, 0%, and 0.6%, respectively. Similarly, on S-NIAH-2 at the same haystack length, Avey attains 68.8% accuracy, whereas Transformer++, Mamba, and RWKV-7 fall to 0%, 2.8%, and 2%, respectively. Comparable trends are observed in the large parameter regime as well, as illustrated in Fig. 7.

An interesting observation arises in the small parameter regime, where Avey outperforms all other models on S-NIAH-1 with a haystack length of 2k, knowing that this length exceeds its trained context window width and enables it to demonstrate its strong extrapolative capability. However, this pattern does not persist in the medium (see Fig. 4 in Section 3.4) and large (see Fig. 7) parameter regimes, where Transformer++ and RWKV-7 outperform Avey on the same benchmark at 2k length, despite this length still surpassing Avey's trained context window. This suggests that these models, with their increased capacity, are able to compensate for the challenge posed by S-NIAH-1, and entails that Avey might benefit from a longer training context window.

In this paper, we kept Avey's context window fixed at 512 tokens across all parameter regimes. All tuning experiments related to sequence length, split size, and top $k$ splits (see Section G) were conducted exclusively using the small model size. It is plausible that with larger capacity, Avey could more effectively leverage longer sequences by retrieving and contextualizing a larger set of relevant tokens while filtering out less informative ones, thereby enhancing contextual representations and further boosting performance. Investigating the relationship between sequence length and model size in Avey is an interesting direction for future work.

## N    COMPLEXITY ANALYSIS

As indicated in Sections 2.1, 2.2.1, 2.2.2, and 2.2.3, the training time complexities of the ranker, enricher, contextualizer, and fuser are $\mathcal{O}(N^2 d)$, $\mathcal{O}(Nmd)$, $\mathcal{O}(NkCm_t)$, and $\mathcal{O}(Nmd)$, respectively, where $N$ is the sequence length, $S$ is the split size, $d$ is the original embedding dimension, $m$ is the projected embedding dimension (with $m > d$), $m_t$ is the tail part of $m$ forwarded to the contextualizer, $C$ is the context width (with $C \leq N$), and $k$ is the number of splits contextualized with each current split. This yielded an overall training time complexity of $\mathcal{O}(N^2 d)$, assuming that scalar multiply-add operations (e.g., those used in computing cosine similarity for MaxSim) and comparisons (e.g., those used to determine maximum scores) are constant-time.

During inference, the time complexities of the enricher, contextualizer, and fuser remain unchanged. However, the ranker's analysis slightly changes, as at *each* time step $t$ (i.e., upon predicting a new token), the current split is compared against all previous splits. More precisely, at each time step $t$, the current split—denoted as split $i$ and incrementally filled as tokens are generated—is compared against all $i - 1$ preceding splits, each consisting of $S$ tokens. Consequently, the cost of comparing $t$ tokens in split $i$ (with $t \leq S$) against $S$ tokens in a previous split is $\mathcal{O}(t \cdot S \cdot d)$.

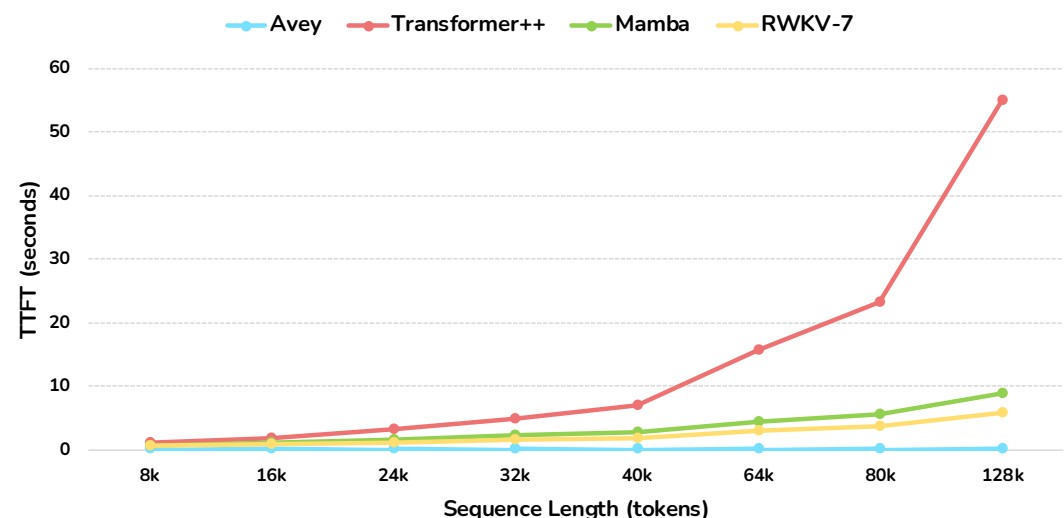

Figure 8: The Time to First Token (TTFT) for Avey, Transformer++, Mamba, and RWKV-7 across varying sequence lengths.

Now, if we let $M = \lceil N/S \rceil \approx N/S$ for large $N$ be the number of splits, the total inference cost can be defined as:

$$\sum_{i=1}^{M}\sum_{t=1}^{S}(i-1)\cdot\mathcal{O}(t\cdot S\cdot d)$$

Simplifying the inner summation yields:

$$\sum_{t=1}^{S}(i-1)\cdot\mathcal{O}(t\cdot S\cdot d) = (i-1)\cdot\mathcal{O}(S\cdot d)\cdot\sum_{t=1}^{S}t = (i-1)\cdot\mathcal{O}(S\cdot d)\cdot\frac{S(S+1)}{2} = (i-1)\cdot\mathcal{O}(S^2\cdot d)$$

Substituting this back into the outer summation gives:

$$\sum_{i=1}^{M}(i-1)\cdot\mathcal{O}(S^2\cdot d) = \mathcal{O}(S^2\cdot d)\cdot\sum_{i=1}^{M}(i-1) = \mathcal{O}(S^2\cdot d)\cdot\frac{M(M-1)}{2}$$

Substituting $M$ with $N/S$ for large $N$ results in:

$$\mathcal{O}(S^2\cdot d)\cdot\frac{(N/S)^2}{2} = \mathcal{O}(S^2\cdot d)\cdot\mathcal{O}(N^2/S^2) = \mathcal{O}(N^2\cdot d)$$

Therefore, the cost per token becomes:

$$\frac{\mathcal{O}(N^2 d)}{N} = \mathcal{O}(Nd)$$

The above analysis indicates that Avey scales linearly during inference. To more precisely characterize Avey's inference-time efficiency relative to other models, we benchmarked Time to First Token (TTFT)—a key latency metric for real-time applications (Horton et al., 2024; Liu et al., 2025; Dexter et al., 2025)— on a single NVIDIA H200 GPU for Avey, Transformer++, Mamba, and RWKV-7 across varying sequence lengths. Figure 8 shows that Transformer++ exhibits an approximately quadratic increase in TTFT as the sequence length $N$ grows, due to its full self-attention mechanism, which operates over the entire prompt before generating the first token in response. In

contrast, Mamba and RWKV-7 scale linearly with $N$, as they require a full forward pass to construct their RNN-style hidden states before emitting the first token. While Avey is also expected to scale linearly in theory, its empirical TTFT values are significantly lower than those of Transformer++, Mamba, and RWKV-7. This discrepancy arises because Avey's dominant contributor to inference complexity, namely, the ranker, is invoked *only once* per forward pass. Consequently, the ranker's computational overhead is minimal in practice, enabling Avey to deliver substantially lower TTFT and making it particularly well-suited for real-world, latency-sensitive applications (e.g., chatbots, and mobile or edge apps).

## O  RELATED WORK

Recurrent Neural Networks (RNNs) (Elman, 1990; Rumelhart et al., 1986) are designed to process sequential data by capturing temporal dependencies, making them well-suited for tasks where input order is essential. However, their cyclical nature limits their potential for parallel computation and exposes them to vanishing and exploding gradient problems. As a result, they typically struggle to effectively learn long-range dependencies. While architectures like Long Short-Term Memory (LSTM) (Hochreiter and Schmidhuber, 1997) and Gated Recurrent Units (GRU) (Cho et al., 2014) mitigate these gradient-related issues, they remain slow to optimize and challenging to scale due to retaining RNN's core recurrent and sequential structure.

In contrast, the Transformer (Vaswani et al., 2017) employs a self-attention mechanism to process each sequence of tokens simultaneously. More precisely, it promotes two key design principles: (1) a recurrent-free architecture, which enables parallel computation of token embeddings, while still capturing their order through positional encodings, and (2) a multi-head self-attention approach, which facilitates cross-token interactions to further enrich the expressiveness of embeddings. These innovations make the Transformer highly effective, as well as parallelizable and efficient to train. However, they also cause its computational and memory requirements to scale quadratically with sequence length, making it expensive and less efficient for very long sequences.

To address the Transformer's quadratic computation and memory costs, a wide array of approaches have been proposed, including linear attention (Kitaev et al., 2020; Katharopoulos et al., 2020; Choromanski et al., 2020; Peng et al., 2021; Zhai et al., 2021), sparse or local attention (Yuan et al., 2025; Child et al., 2019; Parmar et al., 2018), context compression (Rae et al., 2019; Wang et al., 2020; Sukhbaatar et al., 2019; Roy et al., 2021), and modified attention computations (Tay et al., 2021; Wu et al., 2019; Tay et al., 2020), to mention just a few. Notably, the Attention-Free Transformer (AFT) (Zhai et al., 2021) offers a linear drop-in replacement for the quadratic self-attention mechanism. In particular, it weights key and value vectors using learned positional biases and integrates them with query vectors via element-wise multiplication. As such, it eliminates the need to compute and store the costly attention matrix while still preserving global query-value interactions—without requiring architectural modifications or additional tuning. Furthermore, AFT introduces variants such as AFT-local and AFT-conv, which leverage local attention patterns to reduce parameter count and further improve computational and memory efficiency.

RWKV-4 (Peng et al., 2023) (the first 3 versions were experimental (Li et al., 2024b)) capitalizes on AFT and suggests combining the strengths of both Transformers and RNNs. To elaborate, unlike Transformers and akin to RNNs, it does not process each input token solely based on its own embedding, but rather as a weighted sum of its embedding and that of the preceding one. To the contrary of traditional RNNs and similar to Transformers, it adopts self-attention, but an extended version of it, namely, that of AFT. This hybrid approach allows RWKV-4 to maintain some of the modeling capabilities of RNNs, while benefiting from the parallelization and scalability features of Transformers.

More precisely, RWKV-4 extends AFT in two distinct ways: (1) it introduces an additional parameter to handle each current token independently, and (2) it implements a per-time-step decay mechanism that selectively removes older content from the recurrent hidden state in a data-dependent manner. This decay mechanism addresses a central limitation of linear attention, which pertains to the lack of a systematic way to discard outdated information (Schlag et al., 2021; Yang et al., 2023).

Architecturally, RWKV-4 consists of homogeneous stacked residual blocks, each encompassing two units, a time-mixing and a channel-mixing ones. The time-mixing unit applies linear attention across

tokens, while the channel-mixing unit integrates each element of the current token's embedding with its corresponding element from the preceding token embedding, leveraging the output of the time-mixing unit.

RWKV-5 (Peng et al., 2024) enhances RWKV-4's architecture and learning decay mechanism by replacing traditional vector-valued states with more expressive multi-head, matrix-valued ones. Moreover, it reconfigures receptive states, incorporates supplementary gating mechanisms, and dynamically learns the linear interpolation between the current and preceding token embeddings instead of relying on pre-defined hyperparameters.

RWKV-6 (Peng et al., 2024) promotes a new application of low-rank adaptation functions (Hu et al., 2022; Li et al., 2024b). Specifically, it makes the linear interpolation between the current and preceding tokens data-dependent to improve the selectivity of the model in retaining and discarding information. Additionally, it replaces the static decay mechanism with a dynamic one, allowing each element in the decay vector to fluctuate independently over time in response to the input.

The decay strategies of RWKV-4, RWKV-5, and RWKV-6 still cannot remove values stored at specific keys. DeltaNet (Schlag et al., 2021) overcomes this drawback by partially replacing the values stored at current keys with equivalent new values, enabling models to erase outdated memories and include up-to-date ones on a per-key basis. However, it only allows a fixed scalar fraction of a value to be replaced from a state via an in-context learning rate parameter, thus demonstrating rigidity in adapting to varying data contexts.

RWKV-7 (Peng et al., 2025) builds upon the principles of DeltaNet and introduces a vector-valued in-context learning rate instead of a scalar-valued one. This allows selective replacement of state data on a channel-wise basis. Furthermore, RWKV-7 employs a vector-valued decay mechanism and uses additional low-rank projections to optimize the trade-off between the number of parameters, computational efficiency, and downstream performance. Lastly, it incorporates Value Residual Learning (Zhou et al., 2024), which improves the propagation of initial local information via utilizing a residual connection between the value vectors of the current layer and those of the *first* layer prior to the attention operation, resulting in enhanced language modeling performance.

Most recently, RWKV-X (Hou et al., 2025) proposed combining the strengths of RWKV and sparse attention, drawing inspiration from Mixture of Block Attention (MoBA) (Lu et al., 2025). In particular, RWKV-X restricts each query to attend only to a small, relevant subset of the input, thus reducing computational cost and facilitating the modeling of longer-range dependencies. More precisely, rather than allowing each token to attend to every other token in the sequence (as in traditional self-attention), it constrains each token's attention to a limited subset (hence, making it sparse), while maintaining the coupling between the input sequence and context window.

Similar to RWKV, RetNet (Sun et al., 2023) adopts linear attention and promotes a hybrid approach that blends Transformer- and RNN-like representations, yet with a decay-based memory unit. Specifically, it divides the input sequence into chunks, wherein the Transformer-like parallel representation is applied. Additionally, it enables propagating information sequentially across chunks using the RNN-like representation. Lastly, it uses multiple attention heads, each governed by a distinct decay rate, and replaces LayerNorm (Ba et al., 2016) with GroupNorm (Wu and He, 2018).

Although linear attention has been proposed as a promising alternative to quadratic softmax attention (Katharopoulos et al., 2020; Choromanski et al., 2020; Kasai et al., 2021; Peng et al., 2021), existing implementations of it are in practice slower than optimized versions of softmax attention (Dao et al., 2022; Dao, 2023; Yang et al., 2023). From an accuracy standpoint, linear attention generally underperforms conventional softmax attention, sometimes by a significant margin in language modeling (Kasai et al., 2021; Yang et al., 2023).

To this end, and in light of the exponentially growing complexity associated with overcoming the limitations of Transformer-based architectures, there has been a renewed interest in RNN-based alternatives in recent years. Notably, Structured State Space Sequence (S4) models (Gu et al., 2021a;b), inspired by the classical state space models (SSMs) (Kalman, 1960), have emerged as a promising paradigm for sequence modeling. These models describe the temporal evolution of a system using differential equations, offering a continuous-time formulation of dynamics, and can be viewed as generalized versions of RNNs.

An SSM as a concept has a broad meaning, which simply refers to the notion of any recurrent process with a latent state (Gu and Dao, 2023). From this perspective, the RNN-like linear attention model proposed and formulated by (Katharopoulos et al., 2020) can be interpreted as a degenerate linear SSM. Interestingly, this justifies the usage of a decay factor in RetNet and RWKV, especially that a decay term (or a forget gate) has been shown to be crucial in RNNs (Hochreiter and Schmidhuber, 1997; Van Der Westhuizen and Lasenby, 2018; Cho et al., 2014).

Numerous variants of SSMs (Gu et al., 2021a; 2022; Gupta et al., 2022; Li et al., 2024a; Ma et al., 2022; Orvieto et al., 2023; Smith et al., 2022) have demonstrated strong performance across a range of domains, including audio and vision (Goel et al., 2022; Nguyen et al., 2022; Saon et al., 2023). Nonetheless, these variants have struggled with language modeling, often lagging behind Transformers by several points in perplexity (Gu et al., 2021a).

From an efficiency standpoint, however, SSMs have shown encouraging results in language modeling. For instance, S4 (Gu et al., 2021a;b), a prominent SSM, converts the continuous-time state update equation of SSMs into a discrete form, hence, enabling parallel sequence modeling. Moreover, it utilizes the HiPPO (High-Order Polynomial Projection Operator) initialization (Gu et al., 2020), which alleviates the vanishing gradient problem and facilitates processing longer sequences.

Another example of SSMs is H3 (Fu et al., 2022), which improves language modeling by allowing both, the recall of earlier tokens and token-wise comparisons within a sequence. H3 extends S4 by suggesting a state-passing algorithm that enhances computational efficiency on modern hardware. This advancement reduces the hardware-related barriers that have traditionally limited the scalability of SSM-based architectures.

Hyena (Poli et al., 2023) capitalizes on H3 by replacing its S4 layer with an MLP-parameterized global convolution (Romero et al.). S5 (Smith et al., 2022) proposes using parallel scan (Martin and Cundy, 2017) to parallelize S4. Liquid S4 (Hasani et al., 2022) augments S4 with an input-dependent state transition matrix, computed convolutionally in the frequency domain (which is computationally efficient) and mapped back to the time domain using an inverse Fourier transformation. SGConv (Li et al., 2024a), LongConv (Fu et al., 2023), MultiresConv (Shi et al., 2023), and Toeplitz Neural Network (Qin et al., 2023) all focus on the convolutional representation of S4 as well, aiming to enhance its efficiency (Gu and Dao, 2023).

Most recently, Mamba (Gu and Dao, 2023) introduced a new class of SSMs known as *selective* SSMs, specifically designed to improve the performance of language modeling tasks. Mamba addresses a key limitation in SSMs, namely, their inability to selectively process inputs in an input-dependent manner (i.e., focus on or ignore specific parts of the input sequence). Consequently, it makes the SSM parameters input-dependent, but introduces a technical challenge since traditional SSMs are inherently designed to be time- and input-invariant to ensure computational efficiency. To overcome this challenge, Mamba proposes a hardware-efficient parallel scan (or prefix sum) algorithm (Blelloch, 1990), which enables recurrent-style computation without explicitly materializing the expanded state. This design precludes costly I/O operations across GPU memory hierarchies and accelerates both, training and inference.

Tri Dao and Albert Gu (Dao and Gu, 2024) argue that various approaches to operating SSMs can be reframed as matrix multiplication algorithms involving a specific class of structured matrices known as semiseparable matrices. They further leverage the language of tensor contractions to prove the recurrent formulation of linear attention as proposed by (Katharopoulos et al., 2020), before generalizing it to a new family of structured masked attention (SMA).

Subsequently, Tri Dao and Albert Gu demonstrated that SSMs and Transformers are fundamentally connected, governed by the mathematical framework of semiseparable matrices and SMA. Additionally, they developed a rich state space duality (SSD) framework of theoretical connections between SSMs and various forms of attention. This framework facilitated the design of Mamba-2, an extended version of Mamba, which aims to improve its efficiency (not performance). Mamba-2 utilizes a scalar data-dependent gating mechanism (like the ones proposed by (Peng et al., 2021; Sun et al., 2023; Beck et al., 2024)), which enables transforming its recurrent structure into a matrix-multiply form, thus allowing for efficient execution on tensor cores and better support for larger hidden state sizes.

The strategy of the SSD framework mirrors that of linear attention (Katharopoulos et al., 2020), which established a connection between autoregressive attention mechanisms and linear RNNs via showing an equivalence between "dual forms" of quadratic kernelized attention and a specific type of linear recurrence. Conceptually, the SSD framework seeks to transfer algorithmic and systems-level optimizations originally developed for Transformers to the realm of SSMs. Its overarching goal is to enable the development of architectures that outperform Transformers, while scaling more efficiently with sequence length.

Finally, several works, including Tolstikhin *et al.* (Tolstikhin et al., 2021), Melas-Kyriazi (Melas-Kyriazi, 2021), Touvron *et al.* (Touvron et al., 2022), and Ding *et al.* (Ding et al., 2021), among others, have questioned the necessity of self-attention, particularly in the context of Vision Transformers. In contrast, Liu *et al.* (Liu et al., 2021a) introduced gMLP, an MLP-based alternative to BERT-style Transformers (Devlin et al., 2019) that (partially) eliminates self-attention but ultimately underperforms average performance on downstream NLP tasks. gMLP encompasses channel (hidden) and spatial (cross-token) projections with multiplicative gating and *static* parameterization. Its gating mechanism is reminiscent of Gated Linear Units (GLUs) (Dauphin et al., 2017; Shazeer, 2020; Wu et al., 2019), as well as earlier architectures such as Highway Networks (Srivastava et al., 2015) and LSTM-RNNs (Hochreiter and Schmidhuber, 1997). A key distinction, however, is that gMLP applies gating on the spatially projected dimension and not the hidden one. The gated embedding-wise neural network in Avey's contextualizer draws inspiration from gMLP.

Unlike all previously mentioned models, Avey abandons self-attention and recurrence, introducing a new architecture composed of a ranker and a dynamically parameterized neural processor. The ranker identifies the most relevant tokens for contextualization, while the neural processor contextualizes them data-dependently. This design decouples sequence length from context width, enabling efficient processing of arbitrarily long sequences without diminishing the influence of distant yet important tokens.

At the core of Avey's architecture is a weighted-selective-split interaction mechanism, which filters out irrelevant tokens beyond the context window and enables direct interactions only with relevant ones, thus preserving their influence irrespective of sequence length. In addition, Avey employs a partial-embedding bypassing technique that retains a portion of each token's raw, distinctive features before fusing them with its contextualized ones through a neural network. This technique boosts the performance of Avey (as shown in Appendix K) and might help mitigate issues such as entropy collapse (Zhai et al., 2023) and over-smoothing (Zhou et al., 2021; Shi et al., 2022), especially at large-scale, when the depth of the model is increased significantly.

## P    IS THE RANKER A RAG COMPONENT?

The ranker is an *internal* component of Avey that operates *within* the input sequence, selecting among its splits for more effective contextualization. It does not query external corpora or indexes, introduces no retrieval I/O or freshness dependencies, and adds no retrieval latency. Its role is architectural, that is, to allocate Avey's internal contextual budget and decouple context width from sequence length so that Avey can fully contextualize sequences far beyond its training window.

By contrast, Retrieval-Augmented Generation (RAG) (Lewis et al., 2020) augments a model with *external* (non-parametric) knowledge via a retriever, ranging from classic BM25 (Robertson and Zaragoza, 2009) and dense passage retrieval (DPR) (Karpukhin et al., 2020) to system-level designs such as REALM (Guu et al., 2020), RETRO (Borgeaud et al., 2022), and MacRAG (Lim et al., 2025), among others. RAG aims to: (1) improve factuality by grounding outputs in retrieved documents, (2) make models updatable by reflecting new information without retraining, and (3) reduce compute for long-context tasks by moving knowledge out of weights.

As such, the two mechanisms are *orthogonal*. The ranker allocates the model's internal contextual budget over the given sequence, whereas RAG changes the evidence set by importing out-of-sequence content. They can be composed (RAG can be layered atop Avey, as it is with the Transformer) but one does not subsume the other.

## Q   NEURAL CONTEXTUALIZATION VS. ATTENTION

Avey's contextualizer is an embedding-wise neural network that dispenses with attention. In Appendix K, we replaced it with standard self-attention and observed a $4.6\%$ increase in perplexity alongside a $2.1\%$ decline in average task performance, underscoring its central role in Avey's architecture.

Formally, the contextualizer is defined in Equation 2, repeated below for convenience:

$$\mathbf{c}(\mathbf{Z}_t) = \mathbf{Z}_{tl} \odot \sigma\Big(\big(\mathbf{V} \odot \mathcal{N}(\mathbf{Z}_{tr})\mathcal{N}(\mathbf{Z}_{tr})^\top\big)\mathbf{Z}_{tr} + \mathbf{b}'\Big).$$

Let $\mathbf{S} := \mathcal{N}(\mathbf{Z}_{tr})\mathcal{N}(\mathbf{Z}_{tr})^\top$. The product $(\mathbf{V} \odot \mathbf{S})\mathbf{Z}_{tr}$ yields a content-dependent signal that is passed through a pointwise nonlinearity and used to gate $\mathbf{Z}_{tl}$ elementwise, producing a bounded, feature-wise modulation rather than a mixture over values. By contrast, self-attention computes a row-stochastic convex combination of value vectors after a Q/K split and softmax normalization. Evidently, our formulation departs away from both the softmax and the Q/K/V decomposition, whereby weights are neither constrained to be nonnegative nor to sum to one, and the output acts as a gate on carrier features (i.e., $\mathbf{Z}_{tl}$) rather than a convex average of value vectors.

The contextualizer also differs fundamentally from linear attention (Choromanski et al., 2021; Katharopoulos et al., 2020; Wang et al., 2020; Beltagy et al., 2020; Sun et al., 2023). Linear-attention variants obtain near-linear complexity by exploiting an associative kernel factorization that permits reordering and prefix accumulation, typically of the form $\phi(\mathbf{Q})\big(\phi(\mathbf{K})^\top\mathbf{V}_{\text{val}}\big)$. Equation 2 does not admit such reordering. In fact, the Hadamard coupling $(\mathbf{V} \odot \mathbf{S})$ breaks the algebraic associativity required to push multiplications across terms, and the normalization $\mathcal{N}(\cdot)$ is neither linear nor guaranteed nonnegative, precluding the kernel tricks used to approximate softmax attention with associative feature map functions (e.g., ReLU and Exp). Lastly, we note that the contextualizer remains quadratic (not linear) in sequence length.

For similar reasons, Equation 2 cannot be reformulated as a finite-state RNN under an autoregressive mask. Let $\mathbf{S}_t = \mathcal{N}(\mathbf{Z}_{tr}^{\leq t})\mathcal{N}(\mathbf{Z}_{tr}^{\leq t})^\top$. The update at step $t+1$ depends on the full pairwise matrix $(\mathbf{V} \odot \mathbf{S}_t)$, that is, on all position-specific interactions among the past tokens after data-dependent normalization. Because the learned weight matrix $\mathbf{V}$ introduces position-dependent multiplicative couplings, there is no time-invariant transition $h_{t+1} = f(h_t, x_{t+1})$ with a fixed-dimensional sufficient statistic $h_t$ that exactly summarizes $(\mathbf{V} \odot \mathbf{S}_t)$. In particular, the required weights vary across positions and must be recomputed, so any streaming recurrence would either approximate by tying/averaging $\mathbf{V}$ or maintain $\mathcal{O}(t)$ state. Therefore, an exact finite-state RNN equivalence is unavailable.

Empirically, $\mathbf{V}$ performs most of the heavy-lifting in Avey, while $\mathbf{S}$ primarily induces *selectivity*, dynamically emphasizing or suppressing interactions conditioned on the input, echoing the selectivity principle advocated in recent sequence models (Gu and Dao, 2023). An ablation in Appendix K shows that including $\mathbf{S}$ delivers a consistent, albeit modest, gain by making the neural processor's parametrization input-adaptive.

Putting everything together, these distinctions (i.e., gating rather than mixing, non-associative pairwise modulation rather than kernel-factorizable operations, and explicit quadratic interactions), explain both the theoretical departure from self-attention and linear attention and the observed empirical contribution of the contextualizer within Avey.

## R   DESIGN RATIONALE

We designed Avey around clear functional roles for its core modules. Below, we outline some of the guiding intuitions and how they inform its architecture.

**Enricher:**   A substantial body of evidence indicates that much of a language model's knowledge is stored in feed-forward sublayers and accessed through non-linear feature interactions (e.g., (Geva et al., 2021)). The *enricher* is designed accordingly. It serves both as the primary repository of parametric knowledge and as a mechanism for intra-embedding interactions, enabling higher-order,

non-linear composition of features within each embedding. This improves expressivity by allowing features to modulate and refine one another in a context-aware manner.

**Contextualizer:** The *contextualizer* operates as an embedding-wise neural network such that each neuron forms a weighted sum over input embeddings with learned coefficients (see Equation 2). To introduce input-dependent *selectivity* (as in (Gu and Dao, 2023)), we augment these static weights with a cosine-similarity term that produces a second, data-driven set of weights (the two are combined multiplicatively via a Hadamard product). This dynamic modulation improves behaviors such as copying and induction by strengthening interactions that are semantically relevant to the current input. The split-and-gate structure follows established gated designs in gMLP (Liu et al., 2021a) and GLU variants (Dauphin et al., 2017; Shazeer, 2020; Wu et al., 2019).

**Partial Embedding Bypassing:** The enricher's output is partitioned into two streams, one is passed to the contextualizer and the other is bypassed and fed directly to the *fuser*. The bypassed part plays two complementary roles. First, it provides a strong residual path that preserves signal and stabilizes optimization by improving gradient flow within each Avey layer. Second, it supplies additional non-linear capacity in the downstream feed-forward fuser, complementing the contextualizer's primarily linear mixing across embeddings. This balance between context-aware and context-invariant processing yields richer, more diverse representations.

**Fuser:** The *fuser* (a position-wise feed-forward network) learns how to combine the contextualized and bypassed streams and then projects the result back to the model's embedding dimension, ensuring compatibility with residual pathways across layers. As a feed-forward network (FFN), it also contributes to storing and accessing parametric knowledge learned during training, analogous to FFN roles in Transformers.

# S  LIMITATIONS

The scope of our work is limited to textual data and does not involve other modalities such as images, audio, or genomics. Additionally, our evaluation of Avey is restricted to standard autoregressive language modeling, benchmarking it against popular open-source architectures using both pretraining metrics (perplexity) and zero-shot evaluations on established NLP benchmarks. As a result, we do not investigate Avey's ability to construct bidirectional contextualized word representations, as done in BERT (Devlin et al., 2019). We leave this for future work. Finally, the paper focuses solely on effectiveness rather than efficiency. While we provide a complexity analysis showing that Avey exhibits quadratic training time like Transformers, our current implementation is slower. As such, further engineering efforts are required to optimize it.

Table 8: Avey's performance under different sequence lengths, $N$, split sizes, $S$, and top-$k$ values. All models (a total of 69) were trained on 10B tokens using 140 million parameters. The sweet spot in terms of downstream task performance was at $N = 512$, $S = 64$, and $k = 7$; hence, it was adopted as Avey's default configuration.

| $N$ | $S$ | $k$ | Perplexity | ARC-C | ARC-E | HellaSwag | OBQA | PIQA | SIQA | Winogrande | Average |
|---|---|---|---|---|---|---|---|---|---|---|---|
| 256 | 16 | 1 | 43.59 | 21.16 | 37.67 | 30.86 | 27.20 | 64.20 | 37.72 | 51.85 | 38.66 |
| | | 3 | 31.83 | 22.18 | 39.81 | 32.23 | 26.80 | 63.76 | 38.02 | 51.07 | 39.12 |
| | | 5 | 27.14 | 24.15 | 39.02 | 32.57 | 26.00 | 64.20 | 38.02 | 53.20 | 39.59 |
| | | 7 | 25.65 | 22.27 | 38.59 | 31.89 | 28.60 | 64.80 | 37.97 | 50.43 | 39.22 |
| | | 9 | 24.28 | 23.72 | 38.55 | 32.44 | 28.20 | 64.15 | 37.36 | 51.54 | 39.42 |
| | | 11 | 22.89 | 22.53 | 39.27 | 32.45 | 27.40 | 65.07 | 38.02 | 51.30 | 39.43 |
| | | 13 | 23.85 | 22.35 | 38.97 | 31.04 | 26.80 | 64.91 | 37.51 | 49.72 | 38.76 |
| | | 15 | **22.04** | 22.61 | 38.80 | 32.66 | 28.60 | 65.61 | 38.54 | 52.25 | **39.87** |
| | 32 | 1 | 36.58 | 23.98 | 39.18 | 32.57 | 26.80 | 64.25 | 37.36 | 50.83 | 39.28 |
| | | 3 | 31.51 | 22.78 | 39.56 | 33.15 | 25.00 | 64.91 | 38.74 | 52.25 | 39.48 |
| | | 5 | **30.06** | 23.38 | 38.76 | 33.51 | 28.60 | 65.34 | 37.87 | 52.41 | **39.98** |
| | | 7 | 30.58 | 24.23 | 38.97 | 33.14 | 26.80 | 65.23 | 37.82 | 51.14 | 39.62 |
| | 64 | 1 | 32.76 | 23.89 | 39.18 | 33.26 | 27.60 | 66.05 | 38.69 | 50.99 | **39.95** |
| | | 3 | **30.90** | 23.04 | 40.11 | 33.70 | 27.40 | 64.91 | 38.08 | 50.83 | 39.72 |
| | 128 | 1 | **31.30** | 22.27 | 39.65 | 33.07 | 28.20 | 65.45 | 39.20 | 51.78 | **39.95** |
| 512 | 16 | 1 | 42.51 | 22.27 | 37.42 | 31.38 | 27.80 | 63.87 | 37.31 | 51.07 | 38.73 |
| | | 3 | 29.28 | 22.87 | 38.76 | 31.99 | 28.00 | 64.96 | 36.49 | 52.17 | 39.32 |
| | | 5 | 24.64 | 22.01 | 38.05 | 32.69 | 26.80 | 63.98 | 36.44 | 52.01 | 38.85 |
| | | 7 | 23.49 | 23.46 | 38.13 | 31.72 | 26.40 | 64.58 | 37.82 | 49.41 | 38.79 |
| | | 9 | 20.79 | 23.38 | 39.02 | 31.74 | 27.00 | 64.04 | 38.23 | 51.14 | 39.22 |
| | | 11 | 19.52 | 23.63 | 37.88 | 32.13 | 27.20 | 64.91 | 38.02 | 53.67 | 39.63 |
| | | 13 | 19.45 | 22.44 | 37.54 | 31.34 | 26.80 | 63.33 | 36.80 | 50.83 | 38.44 |
| | | 15 | **17.95** | 21.84 | 36.66 | 31.39 | 27.80 | 64.09 | 37.82 | 51.07 | **39.87** |
| | 32 | 1 | 35.47 | 22.87 | 39.94 | 32.49 | 28.40 | 64.47 | 38.18 | 51.70 | 39.72 |
| | | 3 | 29.49 | 22.95 | 39.18 | 33.22 | 25.60 | 65.13 | 39.00 | 50.51 | 39.37 |
| | | 5 | 27.99 | 22.78 | 37.71 | 33.46 | 28.20 | 65.02 | 38.33 | 52.72 | 39.75 |
| | | 7 | 28.07 | 22.01 | 40.07 | 33.45 | 29.20 | 64.80 | 37.67 | 50.75 | 39.71 |
| | | 9 | 27.17 | 23.89 | 39.02 | 33.46 | 28.40 | 64.91 | 38.59 | 50.20 | 39.78 |
| | | 11 | 26.77 | 22.87 | 39.65 | 32.55 | 27.20 | 64.09 | 38.33 | 51.14 | 39.40 |
| | | 13 | **25.72** | 23.55 | 38.97 | 33.52 | 29.00 | 65.67 | 37.56 | 51.85 | **40.02** |
| | | 15 | 26.29 | 22.70 | 39.23 | 32.53 | 29.40 | 64.80 | 38.08 | 50.04 | 39.54 |
| | 64 | 1 | 31.67 | 23.46 | 39.35 | 33.15 | 27.80 | 65.02 | 38.13 | 51.70 | 39.80 |
| | | 3 | 29.51 | 23.38 | 37.92 | 33.12 | 28.40 | 65.72 | 39.10 | 50.83 | 39.78 |
| | | 5 | 29.31 | 24.23 | 39.77 | 33.17 | 27.60 | 64.58 | 38.33 | 52.09 | 39.97 |
| | | 7 | **28.02** | 24.49 | 39.98 | 33.77 | 29.80 | 65.13 | 38.08 | 51.30 | **40.36** |
| | 128 | 1 | **29.25** | 23.72 | 39.90 | 33.76 | 28.20 | 64.09 | 37.10 | 50.99 | 39.68 |
| | | 3 | 29.77 | 22.70 | 39.10 | 33.38 | 28.80 | 65.23 | 38.74 | 51.62 | **39.94** |
| | 256 | 1 | **29.26** | 22.70 | 39.02 | 33.49 | 27.00 | 64.25 | 37.51 | 52.41 | **39.48** |
| 1024 | 16 | 1 | 41.64 | 21.42 | 37.16 | 31.12 | 29.80 | 64.47 | 37.56 | 50.75 | 38.90 |
| | | 3 | 28.08 | 22.61 | 38.26 | 31.88 | 27.20 | 64.64 | 38.08 | 50.99 | 39.09 |
| | | 5 | 23.69 | 22.18 | 38.38 | 31.94 | 28.80 | 64.09 | 37.87 | 51.22 | **39.21** |
| | | 7 | 21.48 | 23.81 | 38.05 | 31.41 | 27.00 | 63.38 | 36.80 | 50.59 | 38.72 |
| | | 9 | 19.83 | 22.53 | 37.50 | 31.90 | 26.80 | 64.47 | 37.92 | 49.64 | 38.68 |
| | | 11 | 18.34 | 21.93 | 37.16 | 31.45 | 28.60 | 65.13 | 37.77 | 50.67 | 38.96 |
| | | 13 | 16.80 | 23.55 | 37.50 | 30.55 | 26.80 | 63.11 | 36.95 | 52.33 | 38.68 |
| | | 15 | **15.33** | 23.29 | 37.54 | 31.04 | 27.60 | 63.06 | 37.77 | 50.91 | 38.74 |
| | 32 | 1 | 35.07 | 22.70 | 39.98 | 32.87 | 27.60 | 65.23 | 37.77 | 51.14 | 39.61 |
| | | 3 | 28.54 | 23.55 | 38.55 | 32.91 | 27.60 | 64.74 | 37.51 | 50.28 | 39.31 |
| | | 5 | 26.25 | 22.95 | 39.06 | 33.39 | 28.60 | 64.64 | 38.28 | 50.04 | 39.57 |
| | | 7 | 26.29 | 24.06 | 38.76 | 32.70 | 27.60 | 64.80 | 37.67 | 53.12 | **39.82** |
| | | 9 | 24.79 | 23.63 | 38.89 | 33.34 | 27.80 | 64.53 | 37.72 | 52.09 | 39.71 |
| | | 11 | 24.33 | 21.93 | 38.76 | 32.56 | 26.40 | 64.36 | 37.36 | 51.38 | 38.96 |
| | | 13 | 23.44 | 22.78 | 37.46 | 32.73 | 29.00 | 65.23 | 37.31 | 50.67 | 39.31 |
| | | 15 | **23.14** | 23.72 | 39.56 | 32.39 | 28.40 | 63.60 | 37.31 | 51.38 | 39.48 |
| | 64 | 1 | 30.84 | 23.89 | 38.51 | 33.31 | 27.20 | 65.18 | 38.28 | 49.88 | 39.46 |
| | | 3 | 27.82 | 22.61 | 39.60 | 33.39 | 28.60 | 64.74 | 38.84 | 50.20 | 39.71 |
| | | 5 | 27.89 | 23.04 | 40.49 | 32.97 | 30.00 | 65.02 | 38.49 | 49.09 | 39.87 |
| | | 7 | 27.48 | 24.06 | 39.27 | 33.54 | 28.80 | 65.72 | 37.72 | 50.75 | 39.98 |
| | | 9 | 27.37 | 22.35 | 39.98 | 33.48 | 28.00 | 65.29 | 38.74 | 52.64 | 40.07 |
| | | 11 | 27.38 | 23.29 | 39.35 | 33.05 | 28.60 | 65.72 | 37.97 | 50.67 | 39.81 |
| | | 13 | 27.44 | 24.23 | 39.77 | 33.05 | 28.20 | 66.05 | 37.31 | 51.07 | 39.95 |
| | | 15 | **26.85** | 24.32 | 40.36 | 34.01 | 28.60 | 65.45 | 37.72 | 51.30 | **40.25** |
| | 128 | 1 | 28.62 | 23.12 | 40.57 | 33.56 | 27.80 | 65.61 | 38.54 | 51.30 | 40.07 |
| | | 3 | **27.08** | 23.89 | 39.81 | 33.65 | 29.00 | 64.69 | 37.72 | 52.09 | 40.12 |
| | | 5 | 28.07 | 23.98 | 40.03 | 33.22 | 29.60 | 65.89 | 39.10 | 50.67 | **40.35** |
| | | 7 | 27.27 | 24.32 | 38.85 | 33.92 | 27.40 | 65.13 | 38.13 | 49.88 | 39.66 |
| | 256 | 1 | **28.30** | 22.61 | 38.38 | 33.22 | 27.80 | 64.85 | 38.79 | 51.30 | **39.56** |
| | | 3 | 28.53 | 23.04 | 39.90 | 32.68 | 27.40 | 64.36 | 37.62 | 49.88 | 39.27 |
| | 512 | 1 | **28.24** | 24.40 | 39.39 | 32.98 | 28.20 | 65.13 | 37.72 | 50.59 | **39.77** |

Table 9: Avey's performance with RMSNorm and LayerNorm. A model with 153 million parameters was trained on 10B tokens using both the neural processor and ranker with the best configuration from Table 8.

| Normalization Method | Perplexity | ARC-C | ARC-E | HellaSwag | OBQA | PIQA | SIQA | Winogrande | Average |
|---|---|---|---|---|---|---|---|---|---|
| RMSNorm | **28.02** | 24.49 | 39.98 | 33.77 | 29.8 | 65.13 | 38.08 | 51.30 | **40.36** |
| LayerNorm | 30.93 | 23.55 | 39.65 | 33.24 | 28.8 | 65.07 | 38.28 | 49.57 | 39.74 |

Table 10: Avey's performance with two types of schedules, constant learning rate (LR) and cosine decay, starting from different peak learning rates. All models involved the neural processor and ranker, and were trained with 153 million parameters on 10B tokens, using both the neural processor and ranker with the best configuration from Table 8.

| Schedule | LR | Perplexity | ARC-C | ARC-E | HellaSwag | OBQA | PIQA | SIQA | Winogrande | Average |
|---|---|---|---|---|---|---|---|---|---|---|
| Constant | 8e-04 | 28.21 | 23.12 | 39.81 | 33.52 | 28.4 | 64.96 | 38.13 | 52.41 | 40.05 |
| | 1e-03 | 27.35 | 24.06 | 40.32 | 33.88 | 29.6 | 65.13 | 38.54 | 51.46 | **40.43** |
| | 3e-03 | 30.38 | 23.81 | 38.34 | 32.90 | 28.2 | 63.38 | 37.77 | 52.33 | 39.53 |
| Cosine Decay | 6e-04 | 26.24 | 22.87 | 40.95 | 33.76 | 29.2 | 65.02 | 37.72 | 49.01 | 39.79 |
| | 8e-04 | 25.64 | 24.06 | 39.31 | 34.43 | 29.6 | 65.83 | 37.87 | 49.80 | 40.13 |
| | 1e-03 | **25.00** | 23.21 | 41.12 | 34.76 | 27.0 | 65.67 | 38.38 | 50.75 | **40.13** |

Table 11: Model configurations used in the scaling law experiments. Each model is trained at three different sizes and numbers of training tokens increased proportionally, following the Chinchilla scaling laws.

| Model | # Layers (# Heads) | Embedding Dim. | Learning Rate | # Tokens |
|---|---|---|---|---|
| Avey-153M | 26 | 768 | 1.00e-03 | 2B |
| Avey-496M | 104 | 768 | 1.00e-03 | 7B |
| Avey-1.5B | 48 | 2048 | 1.00e-03 | 20B |
| Transformer++-152M | 12 (12) | 768 | 3.00e-03 | 2B |
| Transformer++-488M | 26 (16) | 1024 | 1.50e-03 | 7B |
| Transformer++-1.5B | 32 (16) | 1664 | 1.25e-03 | 20B |
| Mamba-153M | 28 | 768 | 3.00e-03 | 2B |
| Mamba-496M | 42 | 1280 | 1.50e-03 | 7B |
| Mamba-1.5B | 52 | 2048 | 1.00e-04 | 20B |
| RWKV-7-152M | 12 | 768 | 6.00e-04 | 2B |
| RWKV-7-488M | 30 | 1024 | 4.00e-04 | 7B |
| RWKV-7-1.5B | 24 | 2048 | 4.00e-04 | 20B |

Table 12: Ablation results comparing Avey variants, with individual components removed or replaced.

| Model Variant | Perplexity | ARC-C | ARC-E | Hella | OBQA | PIQA | SIQA | Wino | Average |
|---|---|---|---|---|---|---|---|---|---|
| Avey full (all features) | 30.00 | 25.17 | 39.90 | 33.59 | 28.8 | 65.56 | 37.62 | 51.62 | **40.32** |
| Avey *without* dynamic parameterization | 34.31 | 25.00 | 40.66 | 32.99 | 28.8 | 65.34 | 36.64 | 50.51 | 39.99 |
| Avey *without* bypassing | 32.55 | 22.61 | 38.38 | 32.31 | 28.0 | 64.20 | 38.28 | 52.09 | 39.41 |
| Avey *without* embedding expansion | 39.94 | 22.44 | 37.92 | 28.75 | 25.4 | 62.40 | 38.64 | 52.01 | 38.22 |
| Avey *without* weighting selected splits | 31.17 | 22.78 | 38.55 | 33.25 | 28.0 | 65.89 | 37.82 | 52.09 | 39.77 |
| Avey *without* the ranker | 29.48 | 23.72 | 38.59 | 32.52 | 28.0 | 63.66 | 37.67 | 53.20 | 39.62 |
| Avey *with* self-attention in place of neural proc. | 31.39 | 22.61 | 39.27 | 31.99 | 28.0 | 64.58 | 38.33 | 51.38 | 39.45 |

Table 13: Performance of all models across short-range benchmarks at 90B, 95B, and 100B training tokens.

| Model (# of Tokens) | ARC-C | ARC-E | HellaSwag | PIQA | OBQA | SIQA | Winogrande | Avg. |
|---|---|---|---|---|---|---|---|---|
| Avey-153M (100BT) | 23.98 | 42.30 | 39.57 | 29.8 | 68.61 | 39.05 | 51.85 | 42.02 |
| Avey-153M (95BT) | 24.23 | 42.09 | 39.21 | 31.2 | 68.23 | 39.15 | 50.28 | 42.06 |
| Avey-153M (90BT) | 24.91 | 42.59 | 39.31 | 33.2 | 68.28 | 39.20 | 51.70 | 42.74 |
| Transformer++-152M (100BT) | 23.29 | 43.43 | 39.47 | 29.4 | 67.03 | 39.10 | 50.51 | 41.90 |
| Transformer++-152M (95BT) | 23.55 | 43.14 | 39.51 | 30.2 | 67.14 | 38.69 | 50.99 | 41.89 |
| Transformer++-152M (90BT) | 24.06 | 42.93 | 38.97 | 29.8 | 66.87 | 38.89 | 49.17 | 41.24 |
| Mamba-144M (100BT) | 24.32 | 43.73 | 40.82 | 29.8 | 68.28 | 39.00 | 52.41 | 42.62 |
| Mamba-144M (95BT) | 23.63 | 43.69 | 40.51 | 32.2 | 68.06 | 39.82 | 53.35 | 43.61 |
| Mamba-144M (90BT) | 24.57 | 43.18 | 40.33 | 29.2 | 68.61 | 39.41 | 52.41 | 42.53 |
| RWKV-7-168M (100BT) | 23.89 | 43.14 | 41.50 | 29.8 | 68.72 | 39.41 | 50.99 | 42.35 |
| RWKV-7-168M (95BT) | 24.23 | 42.89 | 41.77 | 29.2 | 68.99 | 39.10 | 51.14 | 42.48 |
| RWKV-7-168M (90BT) | 24.40 | 43.01 | 41.38 | 30.0 | 68.44 | 39.00 | 51.14 | 42.48 |
| Avey-496M (100BT) | 27.13 | 48.99 | 52.17 | 32.0 | 72.47 | 40.53 | 54.54 | 46.55 |
| Avey-496M (95BT) | 27.90 | 49.20 | 51.74 | 33.0 | 73.07 | 40.63 | 53.51 | 46.72 |
| Avey-496M (90BT) | 27.47 | 48.65 | 51.56 | 32.4 | 71.93 | 39.30 | 55.09 | 46.63 |
| Transformer++-488M (100BT) | 25.68 | 48.02 | 52.92 | 31.6 | 72.69 | 39.56 | 55.96 | 46.06 |
| Transformer++-488M (95BT) | 27.39 | 47.90 | 52.69 | 31.6 | 72.36 | 40.07 | 54.22 | 46.12 |
| Transformer++-488M (90BT) | 27.13 | 48.36 | 52.37 | 32.0 | 71.33 | 40.17 | 55.56 | 46.16 |
| Mamba-500M (100BT) | 29.27 | 51.26 | 54.45 | 34.0 | 73.88 | 40.38 | 54.70 | 48.28 |
| Mamba-500M (95BT) | 28.67 | 51.39 | 54.25 | 34.8 | 72.69 | 40.89 | 55.33 | 48.29 |
| Mamba-500M (90BT) | 27.99 | 50.42 | 53.76 | 34.6 | 72.52 | 41.25 | 56.43 | 48.14 |
| RWKV-7-501M (100BT) | 26.96 | 49.83 | 54.49 | 36.0 | 73.23 | 39.30 | 55.17 | 47.71 |
| RWKV-7-501M (95BT) | 27.39 | 49.24 | 54.66 | 35.6 | 73.78 | 39.15 | 55.80 | 47.95 |
| RWKV-7-501M (90BT) | 27.05 | 49.03 | 54.46 | 37.2 | 73.72 | 39.76 | 56.20 | 48.20 |
| Avey-1.52B (100BT) | 30.89 | 56.36 | 61.49 | 34.4 | 75.84 | 42.07 | 56.59 | 51.09 |
| Avey-1.52B (95BT) | 32.34 | 56.94 | 61.63 | 37.6 | 75.57 | 41.76 | 58.09 | 52.42 |
| Avey-1.52B (90BT) | 30.55 | 56.36 | 61.15 | 38.4 | 75.41 | 42.17 | 56.51 | 51.51 |
| Transformer++-1.5B (100BT) | 30.29 | 56.19 | 64.28 | 38.8 | 76.12 | 42.27 | 61.33 | 52.75 |
| Transformer++-1.5B (95BT) | 30.97 | 57.07 | 63.87 | 37.0 | 76.17 | 42.07 | 61.72 | 52.70 |
| Transformer++-1.5B (90BT) | 28.75 | 55.60 | 63.45 | 38.2 | 75.73 | 42.37 | 61.09 | 52.19 |
| Mamba-1.4B (100BT) | 32.42 | 57.87 | 64.78 | 38.4 | 76.61 | 42.48 | 62.27 | 53.55 |
| Mamba-1.4B (95BT) | 32.85 | 57.91 | 64.37 | 35.4 | 76.33 | 42.02 | 60.93 | 52.69 |
| Mamba-1.4B (90BT) | 32.00 | 58.63 | 64.38 | 36.8 | 76.22 | 41.50 | 61.33 | 52.69 |
| RWKV-7-1.5B (100BT) | 32.42 | 59.55 | 64.59 | 37.4 | 76.82 | 41.86 | 59.67 | 53.19 |
| RWKV-7-1.5B (95BT) | 33.11 | 58.88 | 64.49 | 37.0 | 76.88 | 41.71 | 60.38 | 53.21 |
| RWKV-7-1.5B (90BT) | 33.28 | 58.71 | 64.21 | 37.0 | 76.82 | 41.56 | 60.14 | 53.10 |

Table 14: Summary statistics for each model with different sizes computed over the last three checkpoints (i.e., at 90B, 95B, and 100B training tokens).

| Model | Mean | Standard Deviation | Standard Error | 95% Confidence Interval |
|---|---|---|---|---|
| Avey-153M | 42.32 | 0.3683 | 0.2126 | (41.41, 43.24) |
| Transformer++-152M | 41.72 | 0.1821 | 0.1052 | (41.27, 42.17) |
| Mamba-144M | 42.73 | 0.2700 | 0.1559 | (42.06, 43.40) |
| RWKV-7-168M | 42.48 | 0.0094 | 0.0054 | (42.46, 42.51) |
| Avey-496M | 46.82 | 0.1895 | 0.1094 | (46.35, 47.29) |
| Transformer++-488M | 46.65 | 0.0507 | 0.0293 | (46.52, 46.77) |
| Mamba-500M | 48.23 | 0.0835 | 0.0482 | (48.03, 48.44) |
| RWKV-7-501M | 48.00 | 0.1807 | 0.1043 | (47.55, 48.45) |
| Avey-1.52B | 51.53 | 0.4497 | 0.2596 | (50.41, 52.65) |
| Transformer++-1.5B | 52.54 | 0.3218 | 0.1858 | (51.74, 53.34) |
| Mamba-1.4B | 53.12 | 0.3783 | 0.2184 | (52.18, 54.06) |
| RWKV-7-1.5B | 53.17 | 0.0553 | 0.0320 | (53.03, 53.30) |

