# OpenReview forum: "Don’t Pay Attention"
_ICLR.cc/2026/Conference — ICLR 2026 Conference Withdrawn Submission_

### Official Review · Reviewer_EZM3 · 2025-10-30

**Soundness:** 3
**Presentation:** 2
**Contribution:** 2
**Rating:** 4
**Confidence:** 4

**Summary:**

In this paper, the authors argue that attention-based architectures suffer from quadratic complexity, while recurrence-based architectures have quality issues for long-context modeling. Instead, the authors propose Avey, an attention- and recurrence-free architecture that adopts Ranker to gather relevant tokens from the context thus fixing context length, and Neural Processor to model and predict tokens. In experiments, Avey achieves comparable performance on regular tasks but improved quality on long-context benchmarks.

**Strengths:**

- A new idea of using ranker and processor, rather than traditional attention/recurrence networks, to handle long-context and language modeling.
- Good long-context performance and comparable regular-task results.
- Comprehensive ablations on hyperparameters and design choices.

**Weaknesses:**

- It’s not always clear how the ranker and processor are trained, and how the ranking performance affects the final performance.
- Avey is not attention free: Equation (2) mimics attention.
- Figure 1 could be misleading: while Avey’s context length is 512, its sequence length is not.
- Avey is not scalable: at larger scale, like 1.5B, Avey starts to underperform Transformer++ in Table 2.

**Questions:**

How did you train the ranker? What if just replacing the processor with a vanilla Transformer since the context length is reduced?

---

### Official Review · Reviewer_Fuqi · 2025-10-30

**Soundness:** 1
**Presentation:** 2
**Contribution:** 1
**Rating:** 2
**Confidence:** 4

**Summary:**

The authors introduce a new transformer-like architecture called Avey.

Avey consists of two components: a ranker and a neural processor.  The ranker essentially implements a retrieval-augmented generation (RAG) system.  A long input sequence is broken into "splits" (i.e. chunks), and the ranker will retrieve the Top-K chunks for subsequent operations.  Retrieval uses the MaxSim operation, which returns the maximum cosine similarity between each embedding vector in the current chunk (the chunk that the query vector is in), and the embedding vectors in all other chunks.

The neural processor processes the current chunk, and the chunks retrieved by the ranker.  It implements a variation of transformer attention.

The authors test their architecture on a variety of short-range language modeling tasks, comparing it to Transformers, Mamba, and RWKV.  It performs comparably at small scales, but worse at larger scales.  The authors also test the ranker on a long-range needle-in-a-haystack task, where it performs much better than non-retrieval based models.

**Strengths:**

The paper is clear and relatively easy to follow.  The short-range LM tests are good, covering multiple architectures and tasks.

**Weaknesses:**

The "neural processor" proposed by the authors takes up a large chunk of the paper, and would seem to offer no advantages over any of  the other architectures.  There are numerous problems.  First, it is explained in a confusing way -- the authors do a single projection, and then split it into "head", and "tail", and then further into "tail-left" and "tail-right".  It would be simpler to explain this as three separate projections, like the "keys", "queries" and "values" in a conventional transformer.

Second and more importantly, the "neural processor" is an ad-hoc set of operations that contains analogues of normal transformer attention, but provides no justification as to why the new formulation is any better.  To my eye it look like it should be strictly worse, and experimental results seem to confirm that it is.  There is an attention matrix, this time between "keys" and "keys" (the tail-right projection).  There is a learned relative position matrix V -- comparable to the position bias in T5.  The result is then multiplied by the "values" (tail-left), but there is no softmax, etc.  The authors conduct an ablation and claim that the neural processor is better than normal attention, but this is buried in an appendix, and I would need to see much more extensive experiments before I believed it.

That leaves the ranker as the other main contribution of the paper.  I would first note that the ranker is quite expensive -- it is O(N^2) where N is total context length in tokens, the same as ordinary dense attention.  Most RAG systems compare chunk-level embeddings, rather than token-level embeddings, and thus scale to much larger corpuses.

To my eye, the MaxSim operation seems custom-designed to do very well on needle-in-a-haystack tasks, which is the only task that the authors test on.  I would need to see further experiments, on e.g. coding or long-range Q&A tasks, to convince me that it was a viable replacement.  What's worse, the authors do not actually compare it against other retrievers at all; instead, they compare a retrieval-based architecture against non-retrieval based architectures on needle-in-a-haystack, which is hardly a fair comparison.

In conclusion: the neural processor seems strictly worse than standard attention, and the MaxSim retriever has not been adequately tested.  I think this paper is a clear "reject".

**Questions:**

Have you done any of the following tests?

- Test the ranker + neural-processor   against  ranker + transformer  on long-context tasks.
- Test the ranker against other retrieval mechanisms  (i.e. block-level embeddings).
- Test the neural processor by itself against a normal transformer, without the ranker.

---

### Official Review · Reviewer_Gn8Z · 2025-11-01

**Soundness:** 2
**Presentation:** 2
**Contribution:** 2
**Rating:** 4
**Confidence:** 3

**Summary:**

The authors propose a novel architecture for long-context processing, called Avey. It consists of two main components: a ranker and a neural processor. The neural processor is composed of multiple layers, each containing an enricher, a contextualizer, and a fuser. The authors demonstrate that, even when trained on sequences of only 512 tokens, Avey generalizes effectively to contexts of up to 64k tokens, while achieving slightly lower scores on short-context benchmarks. They also present a comprehensive ablation study analyzing the architectural choices made.

**Strengths:**

- Non-standard architecture choice. Using non-self-attention mechanisms for contextualization and gaining profits from this choice is a valuable point.
- Comprehensive ablation study. The vast majority of architectural choices are supported by ablation experiments.
- Impressive generalization on long sequences in NIAH tasks.

**Weaknesses:**

1. Inductive bias towards NIAH tasks. Using the ranker resembles the use of RAG systems. While being effective on single or few independent needles, it fails when fact relevance depends on the past context. Which leads to the second weakness.
2. Only single-needle NIAH-1 and NIAH-2 were used for long-context evaluation. A model, proposed for long-context processing, should be evaluated on some non-trivial long-context benchmarks. It is unclear how Avey would perform on tasks requiring multiple needles to retrieve information, state tracking, or on tasks requiring sequential hops over needles to reach the answer. Such tasks are present in the RULER, LongBench and BABILong benchmarks (e.g., qa2, qa3), for example.
3. Poor architecture description. A few unclear moments:
- Contextualizer is supposed to be autoregressive. But on Figure 3 arrow points in both directions, which results in confusion. It is either unclear how autoregressive masking is applied, or how parallel training is performed.
- $m_t$ is used both for embeddings and their size in section 2.2.2
- Yellow and Blue colors on different pictures describe different things: on Figure 2 blue color denotes the current split, while on Figure 3 it denotes the fuser output.  Same for the yellow color.
4. Performance drops across almost all short range benchmarks while being unable to solve even NIAH-2 tasks with near-100% quality. The model is designed to solve this type of task.

**Questions:**

questions and comments:

- Could you please elaborate on unclear architecture details in Weaknesses section?
- Abstract (L018) and the main text e.g., (L83-84, L88) mentions "sequence length" and "context width", but do not explain how they differ upon L147. I would suggest briefly describing the distinction between them as early in the text as possible.
- Related Work section is fully moved into appendix. I would suggest to include a concise version in the main paper
- How is Avey related to previous "self-retrieval" works, that retrieve relevant parts from the previous context, e.g. "Retrieval-Pretrained Transformer: Long-range Language Modeling with Self-retrieval" by Rubin, O., 2024?

---

### Official Review · Reviewer_2wSt · 2025-11-03

**Soundness:** 2
**Presentation:** 3
**Contribution:** 2
**Rating:** 4
**Confidence:** 4

**Summary:**

The paper proposes Avey, an attention-free and recurrence-free autoregressive architecture. Avey first ranks equal-sized splits with MaxSim and then contextualizes the current split together with top-k splits via a dynamic, embedding-wise contextualizer

**Strengths:**

Methodological idea is interesting: one-shot, sequence-level selection (ranker invoked once per full forward/backward) followed by non-attention contextualization with a partial-embedding bypass

**Weaknesses:**

1. Novelty: the selection-then-contextualize pipeline resembles prior token/scope reduction themes in sparse Transformers; the novelty mainly lies in (i) MaxSim-based split selection with normalized weighting and (ii) the embedding-wise dynamic contextualizer with bypass.
2. Performance: Short-range accuracy matches a strong Transformer recipe (“Transformer++”). In Table 2 text, the small models show Avey’s average margin +1.43% over Transformer++; with large models Avey is −1.9% on average (no statistical tests reported), suggesting differences may be within typical seed variance for some tasks. This significantly weakens the paper's position. The model neither achieves the linear memory consumption characteristic of linear attention models (like Mamba or RWKV), nor does it demonstrate performance superior to Transformers. And the title is quite overhyped.

**Questions:**

N/A

---

### Note · Authors · 2025-12-03

I have read and agree with the venue's withdrawal policy on behalf of myself and my co-authors.